

**Alternating Extensional and Contractional Tectonics in the**
**West Kunlun Mountains during Jurassic: Responses to the**
**Neo-Tethyan Geodynamics along the Eurasian Margin**
Hong-Xiang Wu[1,2], Han-Lin Chen[1,2*], Andrew V. Zuza[3], Yildirim Dilek[4], Du-Wei
Qiu[1,2], Qi-Ye Lu[1,2], Feng-Qi Zhang[1,2], Xiao-Gan Cheng[1,2], Xiu-Bin Lin[1,2]
1 Key Laboratory of Geoscience Big Data and Deep Resource of Zhejiang
Province, School of Earth Sciences, Zhejiang University, Hangzhou, China.
2 Structural Research Center of Oil & Gas Bearing Basin of Ministry of
Education, Hangzhou, China.
3 Nevada Bureau of Mines and Geology, Nevada Geosicences, University of
Nevada, Reno, NV, USA.
4 Department of Geology & Environmental Earth Science, Miami University,
Oxford, OH, USA.
ORCID: 0000-0003-4997-8715 (Hong-Xiang Wu)
*Corresponding author: hlchen@zju.edu.cn (Han-Lin Chen)
Address: Haina Complex Building No. 1, Zijingang Campus, Zhejiang University,
866 Yuhangtang Road, Hangzhou, 310058, China.



**Abstract:** The Tethyan Orogenic Belt records a long-lived geological cycle
involving subduction and collision along the southern margin of the Eurasian
continent. The West Kunlun Mountains, located at the junction between the
Tibetan and Western Asian Tethyan realm, records multiple orogenic events
from the Paleozoic to the Cenozoic that shape the northwestern Tibetan
Plateau. However, deciphering the complex Mesozoic contractional and
extensional tectonics to interpret the broader Tethyan geodynamics remains
challenging. To address the tectonic transition following the early Cimmerian
(Late Triassic) collision, this study investigates the newly identified Jurassic
sedimentary strata and volcanic rocks in the West Kunlun Mountains. Zircon
geochronological results of basalts and sandstones reveal that this ~ 2.5-km-
thick package was deposited at ca. 178 Ma, rather than the previously reported
Neoproterozoic age. The alkaline basalts at the top of the formation exhibit
chemical compositions similar to oceanic island basalts, consistent with the
intracontinental extension environment revealed by the upward-fining
sedimentary pattern. Provenance analysis, integrating conglomerate clast
lithologies with detrital zircons, suggests a substantial contribution from
adjacent basement sources, likely influenced by the normal faulting during initial
rift stage. These findings indicate that the West Kunlun Mountains rapidly
transitioned into an extensional setting after suturing with Cimmerian terranes.
The regional structure, stratigraphy and magmatism suggest that this Early -
Middle Jurassic basin was subsequently inverted during the Late Jurassic and
earliest Cretaceous. We propose that the Mesozoic deformational history in the
West Kunlun Mountains was related to the northward subduction of the Neo-
Tethys Ocean, as it transitioned from southward retreat to northward flat-slab



advancement. Comparing with the entire strike-length of the Eurasian Tethyan
orogen, we find that the subduction mode varied from the west to the east,
reflecting the broad geodynamic changes to, or initial conditions of, the Neo-
Tethyan system.
**Keywords:** Tethyan Orogenic Belt; West Kunlun Mountains; Jurassic volcanics;
Basin evolution; Subduction retreating and advancing.



## 1 Introduction

The Tethyan Orogenic Belt, a trans-Eurasian mountain system spanning an east-west strike-length of over 15,000 km, is characterized by a series of mountain chains and orogenic plateaus along its latitudinal extent (Fig. 1a; Şengör, 1987; Metcalfe, 2013; Wu et al., 2020). The evolution of the Tethyan Orogenic Belt involved multiple phases of ocean basin opening and closing (i.e., the Proto-, Paleo-, and Neo-Tethys oceans) throughout the Phanerozoic era, which resulted in the development of multiple orogenic belts across the Eurasian continent (Stampfli, 2000; Wan et al., 2019; Metcalfe, 2021). The complex history of accretionary and collisional orogenesis in the Tethyan realm is intricately linked to the breakup and formation of the two mega-landmasses, Gondwana and Laurasia (Şengör et al, 1988; Stampfli and Borel, 2002; Zuza and Yin, 2017; Li et al., 2018; Wang et al., 2018). Documenting the mode and nature of the accretionary and collisional events in the Mesozoic history of the Tethyan orogenic system is, therefore, important for understanding the continental dynamics of Eurasia.

The Mesozoic Tethyan Orogenic Belt involved a protracted phase of orogenesis, rifting, and basin evolution, associated with the convergence between the southern Asian margin and Cimmerian terranes derived from Gondwana (e.g., Kazmin, 1991; Stampfli and Borel, 2002; Angiolini et al., 2013; Robinson, 2015). The tectonic evolution of the Tethyan realm during the Mesozoic exhibits significant variations from the west to the east (Şengör, 1984; Zhu et al., 2022). In the Western Asian section of the Tethyan Orogenic Belt, geochronological and geochemical data from diverse magmatic rocks assemblages suggest a propagating continental rift system in the southern



margin of the Iran Block during the Early Jurassic to Early Cretaceous (Hunziker
et al., 2015; Lechmann et al., 2018; Azizi and Stern, 2019). This process is
envisioned to have been associated with subduction geodynamics involving
multiple intraoceanic subduction zones, slab tearing, and alternating slab
rollback and advance within Neo-Tethys (Zhang et al., 2018; Jafari et al., 2023).
Conversely, in the Eastern Asian section of the Tethyan Orogenic Belt (i.e.
Tibetan sector), an Andean-type orogeny along the southern margin of Eurasia
from the Early Jurassic to the Early Cretaceous has been proposed to explain
deformation and sedimentation patterns in the southern Tibetan Plateau (Kapp
et al., 2007; Zhang et al., 2012; Xie and Dilek, 2023). This process was
punctuated by Toarcian-Aalenian back-arc rifting event resulting from retreat of
the subducting Neo-Tethyan seafloor (Hou et al., 2015; Wei et al., 2017).
The West Kunlun Mountains, stretching from the northern Pamir to
northwestern Tibetan Plateau, occupy a critical position at the junction between
the western and eastern Tethyan Orogenic Belts (Fig. 1b; Şengör, 1984; Wu et
al., 2016). The Kunlun Mountains involved the closure of the Paleo-Tethyan
Ocean in the Triassic-Jurassic, followed by Cenozoic deformation and uplift
during the Himalayan orogeny (Mattern and Schneider, 2000; Cao et al., 2015;
Li et al., 2019; Xiao et al., 2002). Hence, the Mesozoic geology of the West
Kunlun Mountains documents the plate tectonic history of the junction region
within the Tethyan realm, providing pivotal insights into the formation of this
extensive orogenic system. In particular, the Cimmerian Orogeny in the West
Kunlun region critically represents the collision between the Gondwana- derived
continental fragments and the southern Eurasian margin in the latest Triassic
to late Jurassic (e.g., Şengör, 1979), but the timing and duration of this orogen



remains equivocal. Existing interpretations of the Jurassic palaeogeography
and evolution vary, ranging from syn-orogenic (Cao et al., 2015), post-orogenic
(Wu et al., 2021), to transtensional (Sobel, 1999), because of the scarcity of the
relevant geological record from this period. Significant challenges also persist
in understanding the Mesozoic evolution of the Pamir terranes (Angiolini et al.,
2013), including the timing of suturing and exact kinematics of related
deformation (Robinson, 2015). The Cenozoic contractional deformation
episodes, due the northward subduction of the Neo-Tethyan Ocean and the
collision of India with Eurasia, further complicates our understanding in this
remote region (Burtman and Molnar, 1993; Cowgill, 2010). The limited
knowledge of the Jurassic and Cretaceous evolution of the Pamir interior has
been preliminarily deduced from the timing and nature of regional magmatic
activities (Chapman et al., 2018) that are challenged by the information derived
from the surrounding, fragmented sedimentary basins (Leith, 1985; Wu et al.,

2021).

To better understand the regional evolution and tectono-magmatic

processes in the West Kunlun Mountains, we have undertaken a systematic
geochronological and geochemical study and detailed analyses of sedimentary
provenance of volcaniclastic rock suites in a Jurassic basin. By integrating
these new results with existing data from the adjacent region, this study
provides further constraints on the Mesozoic tectonic history of the central
junction of the Tethyan Orogenic Belt, probing the preceding processes that
cause the formation of the broad plateau in central Asia.



## 2 Geological framework and sampling

### 2.1 Tethyan history



The Tethyan Orogenic Belt is a vast, east-west-extending mountain system
that separates the main Eurasian cratons and stable platforms in the north from
Gondwana - derived continental terranes in the south (e.g., Şengör et al, 1988;
Stampfli et al., 1991). The development of the Tethyan Orogenic Belt involves
the evolution of multiple ocean basins and their seaways, including the Proto-
Tethys, Paleo-Tethys, and Neo-Tethys (Stampfli, 2000; Metcalfe, 2021). These
ancient ocean basins overlapped in time but closed successively as the
Gondwana - derived ribbon continents (i.e., Apulia, Pelagonia, Sakarya, Tauride,
and Lhasa) accreted to the southern margin of Eurasia, creating several sub-
parallel suture zones stretching from the circum-Mediterranean region,
Caucasus, Iranian Plateau, and continuing eastward into the Tibetan Plateau
and Southeast Asia (Fig. 1a; Dilek and Moores, 1990; Wu et al., 2020; Metcalfe,

2021).

The Cenozoic indentation of the Pamirs fundamentally affected the
deformation pattern of the Tethyan Orogenic Belt and geographically divided
the belt into western and eastern sectors (Tapponnier et al., 1981). The history
of the Proto-Tethys was linked to the breakup of the Rodinia supercontinent
(Zhao et al., 2018). The western segment of the Proto-Tethys has been defined
as a Cambrian-Silurian ocean existing between Baltica and Gondwana,
whereas the eastern Proto-Tethys appears to have been closed earlier in the
Early Silurian, as a series of Asian blocks collided onto the northern margin of
Gondwana (e.g., Stampfli and Borel, 2002). The opening of the Paleo- and Neo-
Tethyan ocean basins was related to slab pull forces that caused the



detachment of the Hun (including the Tarim, North and South China) and
Cimmerian terrane ribbons from the northern margin of Gondwanaland,
respectively (Stampfli and Borel, 2002; Ruban et al., 2007). These terranes
were successively transferred northward to the Eurasian continent, causing the
closure of these internal seaways during the Cimmerian and Himalayan
orogenies at the end of the Triassic and the beginning of the Cenozoic,
respectively (Dilek and Furnes, 2019; Wan et al., 2019).

The final demise of the Paleo-Tethyan Ocean and the initiation of

subduction in the Neo-Tethyan Ocean occurred simultaneously in the Triassic -
earliest Jurassic, which is of vital importance for comprehension of the cyclical
Tethyan evolution (Wan et al., 2019). The West Kunlun Mountains, situated to
the north of the Pamir syntaxis, forms the western extent of the Tibetan Plateau
(Fig. 1b-c). They constitute an important spatial link between the western and
eastern domains of the Tethyan Orogenic Belt. The formation of the West and
East Kunlun Mountains, involved accretionary and collisional orogeneses
during the closure of the Proto-Tethys and Paleo-Tethys oceans (Mattern and
Schneider, 2000; Xiao et al., 2005; Dong et al., 2018). The East Kunlun
Mountains are deflected to the north relative to the West Kunlun Mountains by
the dextral Altyn-Tagh strike-slip fault (Fig. 1b). During the Early Paleozoic, the
closure of the Proto-Tethys Ocean led to the collision of the Tarim Craton (North
Kunlun) and the South Kunlun terrane along the Kudi suture zone (Fig. 1c;
Zhang et al., 2019a). After splitting from eastern Gondwana in the Devonian -
Carboniferous, the Tianshuihai - Qiangtang blocks travelled northward towards
the Tarim Craton because of the subduction of the Paleo-Tethyan Ocean floor.
These blocks ultimately collided with the Tarim Craton at the latest Triassic,





forming the Mazar - Kangxiwa suture zone (Fig. 1c; Xiao et al., 2005; Metcalfe,
2021). The Pamir terranes (including the Central Pamir, South Pamir, and
Karakoram), commonly regarded as the western counterpart of the Qiangtang
block, rifted from Gondwana much later, during the Permian (Robinson, 2015;
Angiolini et al., 2015). The major Cimmerian orogenic unconformity between
the Lower Jurassic and the deformed Upper Triassic strata is generally
considered to mark the timing of the integration of these Pamir terranes onto
the Eurasian margin (Angiolini et al., 2013; Li et al., 2022b).

The mid-Mesozoic tectonic evolution of the West Kunlun Mountains and

Pamir is somewhat enigmatic, as the first-order geodynamic mechanisms for
widespread observed deformation remain unclear. The interpretation of
Jurassic molasse deposits has led to differing understandings on the tectonic
setting in the region, such as syn-orogeny or post-collisional rifting (Gaetani et
al., 1993; Wu et al., 2021). Several major exhumation events, including the Late
Triassic and Early Jurassic, Middle-Late Jurassic, Early Cretaceous, and Late
Cretaceous, are documented by low-temperature thermochronology in the
mountain ranges and surrounding basins (Sobel, 2013; Cao et al., 2015; Li et
al., 2019, 2023). Mid-Cretaceous granitoid plutons are widespread in the South
Pamir and Karakoram. A polymetamorphic Jurassic and Cretaceous history of
the mountains is also displayed by monazite ages (Faisal et al., 2014). The
basement cooling as well as magmatic, and metamorphic events have
previously been interpreted as associated with far-field stress effects of
collisional events (Yang et al., 2017) or a high-flux event during an Andean-type
subduction of the Neo-Tethyan Ocean (Chapman et al., 2018). These Mesozoic
structures within the orogenic belts were intensely reworked by the Cenozoic



deformation during the Himalayan orogeny (Burtman and Molnar, 1993).

**2.2 Regional geology and sampling strategy**

This study focused on the central and southern parts of the northwest-
trending Jurassic basin within the West Kunlun Mountains (Fig. 1c). The
Kyzyltau region, situated in the central part of this Jurassic basin, preserves the
thickest Early-Middle Jurassic strata. It mainly comprises the Lower Jurassic
Shalitashi and Kangsu formations, and the Middle Jurassic Yangye and Taerga
formations (Fig. 2a). The Shalitashi Formation comprises a massive, thick
conglomerate that overlies the deformed Carboniferous and Permian shallow
marine clastic rocks and limestones along an angular unconformity (Fig. 3a).
The poorly sorted textures and lateral thickness variations in the conglomerate
indicate that its clastic material originated from alluvial fans (Sobel, 1999; Fig.
3b). The Kangsu and Yangye formations form the main part of the Jurassic
strata (Fig. 2a), with total stratigraphic thickness exceeding 1800 meters. The
Kangsu Formation mainly comprises stacked greywackes interbedded with
coal layers. The Yangye Formation consists mainly of interbedded sandstones
and shales exhibiting typical Bouma sequences, indicative of turbidite deposits
in a deepwater environment (Wu et al., 2021). The Middle Jurassic Taerga
Formation is only exposed in the northeastern side of the region and consists
of thinly-bedded shales and siltstones. The Lower to Middle Jurassic
stratigraphy forms an upward-fining sequence, indicating the expanding and
deepening of the basin over time. Structurally, the Jurassic strata exhibit strong
deformation, forming a northwest-trending synclinorium (Fig. 2a). The Cenozoic
contraction in the region extensively deformed the coal-bearing strata, resulting





232 in the formation of multi-scale folds and thrusts (Fig. 3c and 3d). Regionally, the

233 Early-Middle Jurassic strata are unconformably overlain by the Late Jurassic

234 Kuzigongsu Formation and the Cretaceous Kezilesu Group, which are

235 characterized by oxidation-colored, massive conglomerate and sandstones

236 (Fig. 3e). This event was generally interpreted to have been linked to the Middle

237 - Late Jurassic, large-scale contraction and aridification across central Asia

238 (Hendrix et al., 1992; Yang et al., 2017).

239  Documentation and study of the Mesozoic stratigraphy in the southernmost

240 part of the Jurassic basin have been relatively insufficient. In the Kandilik region,

241 geological mapping identified a coal-bearing formation, known as the Lower -

242 Middle Jurassic Yarkant Formation, and a massive conglomerate classified as

243 the Upper Jurassic Kuzigongsu Formation (Fig. 2b). These Jurassic strata were

244 strongly deformed and laterally bounded by a mylonitic shear zone to the west

245 and thrust faults to the east. A stratigraphic unit of gray-black slate interbedded

246 with fine sandstones and siltstones is exposed to the east of the Yarkant

247 Formation, with a thickness exceeding 3500 meters (Ma et al., 1991). Abundant

248 mafic dykes intruded into the lower part of the strata (Fig. 3f), causing local

249 contact metamorphism. A suite of volcanic strata composed of several basalt

250 layers are juxtaposed with the thick clastic package along a steeply-dipping

251 fault. Several eruptive episodes are identified within this unit based on

252 alternating volcanic horizons, including volcanic breccia (Fig. 3g), amygdaloidal

253 basalts, and massive basalts (Fig. 3h). These volcanic rocks belong to the part

254 of upper member deposited above the thick clastic strata (Ma et al., 1991). Due

255 to the lack of reliable constraints from chronological results, this stratigraphic

256 unit has long been thought as Precambrian in age (Ma et al., 1991). Structurally,



the strata were intensely deformed by regional Kashgar-Yecheng transfer faults
(Fig. 2) and bedding dips steeply to the northeast (Fig. 3i).

In the Kandilik region, one basalt sample (AYBL09) was collected near the

thrust fault for geochronological dating (Fig. 2b). Six fresh, undeformed basalt
samples were also obtained away from faults for geochemical analysis. These
basaltic rock samples consist primarily of plagioclase with a fine columnar
texture and anhedral Ti-Fe oxides (Fig. 3j). Plagioclase is locally altered into
chlorite. Additionally, one quartz-lithic sandstone sample (AYBL13) was
collected for detrital zircon age analysis. This sample exhibits poor sorting and
is composed mainly of quartz (~ 30%) with angular shapes, feldspar (<10%),
and lithic fragments (> 60%) (Fig. 3k). For regional comparison, two sandstone
samples were collected from the Kangsu (KZLT1601) and Yangye formations
(KZLT1602) in the Kyzyltau region (Fig. 2a). These sandstones show similar
textures and compositions to the clastic sample from the Kandilik region (Fig.
3l).

**3 Methodology**

One basalt sample (AYBL09) was collected from the Kandilik region for

zircon U‐Pb geochronology and in-situ trace element analysis. Zircon
separation and cathodoluminescence (CL) imaging were done at Yuheng Rock
& Mineral Technology Service Co., LTD., Langfang, China. Zircons were
analyzed for U‐Pb geochronology using an Agilent 8900 ICP-QQQ equipped
with an ESI New Wave NWR 193UC (Two Vol2) laser ablation system at Beijing
Quick-Thermo Science & Technology Co., Ltd, China. Concordia plots were
constructed using IsoplotR (Vermeesch, 2018).



To analyze the petrogenesis and tectonic setting of magmatism, six fresh
basalt rocks were collected from the same section for determining their major
and trace element chemistry. Samples were first crushed, and powdered in an
agate mill. Elemental analyses were conducted at Wuhan SampleSolution
Analytical Technology Co., Ltd. Major-element analyses were performed by X-
ray fluorescence spectrometry (ZSXPrimusII), with analytical uncertainties
generally better than 1%. Trace-element contents were determined using an
Agilent 7700e ICP-MS.
To compare the detrital age patterns and sedimentary provenance, we
have conducted zircon U-Pb dating on two sandstones (KZLT1601 and KZLT1602)
exposed in the Kyzyltau section, and one sandstone (AYBL13) exposed in the
Kandilik section (Fig. 2B). Zircons from samples KZLT1601 and KZLT1602 were
analyzed for U‑Pb geochronology using a Thermofisher iCAP RQ ICP-MS
equipped with a Cetea Analyte HE laser ablation system at School of Earth
Sciences, Zhejiang University. Zircons from sample AYBL13 were analyzed for
U‑Pb geochronology using an Agilent 8900 ICP-QQQ equipped with an ESI
New Wave NWR 193UC (Two Vol2) laser ablation system at Beijing Quick-
Thermo Science & Technology Co., Ltd. The Common Pb was corrected with
the method proposed by (Andersen, 2002). Concordia plots and Kernel Density
Estimate (KDE) plots were constructed using IsoplotR (Vermeesch, 2018) and
Density Plotter 8.5 (Vermeesch, 2012), respectively.
The details of the analytical procedures and the information of the
analytical methodologies, as explained above, are presented in Table S1.
The data from the conglomerate in the Shalitashi Formation were collected
at eight different sections. Analysis of conglomerate clasts was conducted





within a designated 1 square meter area. Our focus was on documenting the
lithological compositions of the clasts, with at least one hundred gravels
randomly counted at each site.

**4 Analytical Results**
**4.1 Morphology and geochronology of zircons from basalt samples**
The results of zircon U-Pb dating of the basalt sample are presented in
Table S2. Approximately one hundred and seventy zircon grains have been
successfully separated from the basalt sample. Zircon crystals are mostly
transparent and colorless, displaying varying lengths ranging between 50-200
μm with elongation ratios of 1:1-5:1 (Fig. 4). Upon examination of their
cathodoluminescence (CL) images, we have sub-categorized these zircons into
two groups based on the presence of oscillatory zoning. The grains showing
well-defined growth zoning are generally sub-euhedral in shape (no.3 in Fig. 4),
which imply their magmatic origin (Fig. 4; Hoskin and Schaltegger, 2003).
Another type of zircon displays inconspicuous zoning texture or yields only
faintly visible zoning patterns (no.15 in Fig. 4). Morphological analysis of these
zircons reveals a range from needle-shaped and elongated crystals (no.13 in
Fig. 4) to stubby and equant forms (no.12 in Fig. 4). A common feature of these
varying grains is their subrounded external appearance. This may result from
moderate resorption either during the evolution of the magma chamber when
the magma is oversaturated with respect to zircon or a certain degree of
metamorphism (Corfu et al., 2003). In addition to their "polished" shape, these
zircons commonly display nebulous or patchy-zoned centers, without distinct
core-rim structures (no.11-13 in Fig. 4).



We have conducted a total of thirty-six spot analyses on various types of
zircons, resulting in thirty-three analyses with a > 90% concordance (Fig. 5a).
The Th/U ratios of these zircons range from 0.04 to 1.52 (Fig. 5d). We cannot
assert that all of them are primary crystals without modification simply based
on the evaluation of Th/U ratios. However, all of these results yielded
concordant ages spanning a broad range from the Early Neoproterozoic to the
Jurassic. Twenty youngest zircons with the concordant ages define a weighted
mean $^{206}Pb/^{238}U$ age of 178±2 Ma (MSWD = 0.99) (Fig. 5b). We interpret this
Toarcian age as the crystallization age of the zircons in this rock sample. The
remaining older zircons yield primarily middle Paleozoic and Neoproterozoic
ages, which we interpret as inherited from the country rock.

**4.2 Detrital zircon U–Pb ages from Jurassic sandstone**

The zircon U-Pb geochronological dataset for the detrital zircons is
presented in Table S2. A total of 101 spot analyses were conducted on zircon
grains from sample AYBL13. After filtering grains with greater than 10% age
discordance, 98 of them met the criteria for inclusion in the Kernel Density
Estimate (KDE) visualization (Fig. 6a). The analyzed results reveal that the
Th/U ratios of most effective zircons range between 0.12 and 2.61, with only
four zircons yielding extremely low values below 0.1 (Fig. 5d). The results
suggest that most detrital zircons from sample AYBL13 are of igneous origin
(Belousova et al., 2002). The youngest zircon grain from this sandstone yielded
an apparent $^{206}Pb/^{223}U$ age of 429 ± 5Ma, whereas the oldest grain has
revealed an apparent $^{206}Pb/^{207}Pb$ age of 3080 ± 22 Ma. The KDE plot reveals
four main age populations with peaks at approximately 446 Ma, 820-955 Ma,





1553 Ma, and 2484 Ma (Fig. 6b).
For analyzing regional detrital provenance, two Jurassic samples from
Kyzyltau were analyzed for age comparison. The Early Jurassic sample
KZLT1601 underwent one hundred spot analyses on randomly selected zircon
grains. These measured grains exhibit Th/U ratios ranging from 0.09 to 1.49
(Fig. 5d), consistent with an igneous origin. Eighty-nine zircon ages were
plotted on or near the concordant curve (Fig. 6c), providing zircon ages ranging
from 369 ± 6 Ma to 3314 ± 15 Ma. The detrital age spectrum was obtained using
the KDE method and revealed similar peaks at approximately 444 Ma, 807 Ma,
1823 Ma, and 2566 Ma (Fig. 6d).
Similarly, one hundred zircon grains from the Middle Jurassic sample
KZLT1602 exhibit characteristics indicative of a magmatic origin, with high Th/U
ratios ranging between 0.11 and 2.63 (Fig. 5d). Ninety - eight concordant results
display consistent age population with the sample KZLT1601, ranging from 345
± 4 Ma to 3029 ± 15 Ma (Fig. 6e). These age populations on the KDE plot also
display four main peaks at approximately 435 Ma, 782-988 Ma, 1829 Ma, and
2480 Ma (Fig. 6f).

**4.3 Analysis of Jurassic conglomerate clast lithologies**
The field provenance analysis of the Lower Jurassic conglomerate
(Shalitashi Formation) reveals significant variations in composition across
different sections. In the Kangsu and Wulagen sections, located in the
northernmost region of the West Kunlun Range, clasts are composed
predominantly of green sandstones (80-51%) and low-grade metamorphic
rocks like schist (0-46%), with minor occurrences of light-colored siliceous rock



(14-3%) and granitoid (6-0%). In the northwestern sector of the Pamir, a
variegated sandstone (22-46%) and a recycled siliceous rock (29-46%)
predominantly constitute major clasts in the Oytag and Gaizi sections,
respectively. Additionally, minor limestone (11-2%) and diverse igneous rocks
(38-6%), including granitoids, rhyolite, and basalts occur characteristically in
the same stratigraphic horizon. In the Kyzyltau section, the clasts of the
Jurassic conglomerate are dominated by green-colored sandstone (28%) and
granites (50%) with subordinate schist (13%) and siliceous rock (9%). To the
south of Kyzyltau, the Tamu and Qimugen sections present a provenance
source dominated by sedimentary rocks. Clasts of limestone and green
sandstone account for 85% and 61% in the neighboring sections, respectively.
The proportion of reddish sandstone in the Qimugen section (33%) surpasses
that in the Tamu section (15%). The Kusilafu section, located to the north of the
Kandilik region, exhibits similar clast lithologies in the conglomerate to the
Qimugen section, with a predominance of green sandstone (34%) and recycled
siliceous rock (45%), along with minor occurrences of reddish sandstone (16%).
Detailed clast lithologies and counting results are presented in the Table S4.

**4.4 Whole-rock major and trace elements of basalts**

The chemical compositions of the basalt samples from the Kandilik section
are provided in Table S5. Except for one sample (AYBL11D), the majority of our
samples displays similar geochemical compositions, characterized by low $SiO_2$
(45.7-51.0 wt.%) and MgO (4.78-7.18 wt.%) contents, and Mg#s ranging
between 45 and 52. These samples possess high $TiO_2$ (2.42-3.34 wt.%) and
total alkali ($Na_2O + K_2O$ = 5.17-6.35 wt.%) contents, exhibit moderate $Al_2O_3$





contents ranging from 11.1 to 14.4 wt.% and total $Fe_2O_3$ ranging from 12.6 to
13.7 wt.%. In comparison, the sample AYBL11D displays relatively high
contents of $SiO_2$ (55.5 wt.%) and $TiO_2$ (4.76 wt.%) with a low total alkali content
(4.80 wt.%). All basalt samples fall within the alkaline series field as depicted in
the total alkali-silica diagram (Fig. 7a). However, it is worth noting that all
analyzed samples exhibit varying Lost-on-Ignition (LOI = 1.51-9.81 wt.%)
values, attributed to weathering and alteration effects, with the presence of
chlorite and calcite (Fig. 3j). Hence, it is crucial to assess the alteration effects
on the chemical compositions of the analyzed samples. The high-field-strength
elements (HFSE, such as Nb, Ta, Ti, and Hf) and rare earth elements (REE)
are typically immobile during alteration. This is supported by the consistent
elemental variations against the most immobile element Zr, as shown in the Fig.
S1. Additionally, Cr and Ni in these samples (except AYBL11D) also
demonstrate strong correlations with Zr, suggesting that these elements were
essentially immobile during alteration. Based on the Nb/Y vs. $Zr/TiO_2$ diagram
proposed by Winchester and Floyd (1977), all samples plot in the alkaline series
(Fig. 7b). Therefore, we posit that these rocks are best classified as alkaline
basalt.

All analyzed samples display consistent chondrite-normalized rare earth

element patterns (Fig. 7c), characterized by an enrichment of LREE relative to
HREE, with $(La/Yb)_N$ ratios ranging from 6.24 to 7.96. Moreover, their REE
patterns exhibit slight negative Eu anomalies (δEu = 0.7-1.0). The primitive
mantle-normalized multi-element diagram illustrates that the analyzed samples
are characterized by the enrichment of highly incompatible trace elements
relative to low incompatible elements (Fig. 7d). The samples present significant



depletion of Sr and slight enrichment in Zr and Hf. No negative Zr-Hf-Ti
anomalies are observed in any of the analyzed basalts.

**5 Identification and age constraints for the Lower Jurassic strata**

Identified Jurassic strata are largely exposed in the eastern edge of the
West Kunlun Mountains and on the southern side along the Talas-Fergana
Fault (Fig. 1c). The Jurassic sequences are comprised of coal-bearing
siliciclastic rocks with variable thicknesses (Wu et al., 2021). Jurassic volcanic
strata have not been previously identified in the West Kunlun Mountains,
although a Jurassic tuffaceous succession and Upper Triassic - Lower Jurassic
volcanic rocks crop out in the Hindu Kush along the western edge of the Pamir
(Brookfield and Hashmat, 2001). Our study has focused on a package of thick
clastic rocks intercalated with basaltic lavas, are exposed in the southermost
part of the Jurassic Kyzyltau syncline (Fig. 2). This stratigraphic package was
previously considered to be of Mesoproterozoic or Neoproterozoic age due to
the lack of fossil records and the presence of low-degree metamorphism (Ma
et al., 1991). Lithologically, the monotonous clastic member is composed
primarily of gray-black slate and fine - grained sandstone to siltstone, rich in
iron and carbonaceous components (Ma et al., 1991). The overlying basalts
vary significantly in their thickness and lithological makeup, composed primarily
of basaltic volcanic breccia, amygdaloidal, and massive layers (Fig.3g and 3h).
Our new results of zircon U-Pb dating of basalts and sandstones suggest
that this rock assemblage is not Precambrian in age, given the widespread
appearance of Phanerozoic ages. We suggest that the weighted mean
$^{206}Pb/^{238}U$ age (~178 Ma) of the youngest group of zircons separated from the





basalt sample could define the eruptive age of this magmatic episode based on
the following lines of evidence. First, these zircons exhibit similar morphological
and CL imaging characteristics (Fig. 4), with the majority of the analyzed grains
displaying Th/U ratios indicating their igneous origin (Fig. 5d). Secondly, the
results of our in-situ trace elemental composition of the zircons (Table S3)
indicate that the chondrite-normalized rare earth elements consistently exhibit
left-sloping pattern with positive anomalies in Ce and Sm, and negative
anomalies in Eu, similar to those of typical igneous zircons (Fig. 5c; Hoskin and
Schaltegger, 2003). Thirdly, according to the Y vs. Yb/Sm plot proposed by
Belousova et al. (2002), these Jurassic zircons are consistent with the basic or
ultrabasic igneous origin (Fig. 5e). Thus, we posit that the crystallization age of
the basalt is Toarcian.
To refine the depositional age of the clastic member of the stratigraphy, we
have compared the detrital zircon results from the feldspar lithic sandstones
with those from the Lower and Middle Jurassic strata, exposed in the Kyzyltau
region. The sandstone collected from the Kangsu Formation displayed similar
texture and composition to the rocks from the Kandilik region, both composed
of immature and poorly sorted quartz and lithic fragments (Fig. 3k and 3i). The
age patterns of detrital zircons display remarkably similar populations with Early
Silurian (~440 Ma) and Tonian (~800-950 Ma) dominated peaks, indicating that
sediments of the two investigated areas shared a common exhumed
provenance. The Lower and Middle Jurassic sedimentary rocks were previously
suggested to have been deposited within structural half grabens and mostly
sourced from the West Kunlun Mountains (Chen et al., 2018). This
interpretation is consistent with our findings. Furthermore, we infer that this





stratigraphic package resembles a turbidite sequence, exhibiting relatively
proximal, deep-water depositional features.
Accordingly, we propose reassigning this thick package of clastic rocks to
the Early - Middle Jurassic age. Hereon, we demonstrate the structural
compatibility of this new stratigraphic scheme. The Lower - Middle Jurassic
strata of the Yarkant Formation in the studied region comprise a lacustrine
association rich in coal beds, and it delineated structurally by a mylonite zone
to its west (Fig. 2b). The redefined sequences are rich in carbonaceous
components and are closely bounded by Jurassic coal-bearing strata along
several reverse faults. These two units successfully extend into the NW-SE-
striking Jurassic graben, which surprisingly narrows rapidly towards the south
without any obvious facies transition (Fig. 1c). The basin-ward dipping of the
strata constituted the western limb of the Jurassic syncline, which has a
comparable thickness that may extend into the southern area of the Kyzyltau
syncline (Fig. 2).

**6 Discussion**
**6.1 Generation and geological setting of the Early Jurassic volcanism**
The basalt samples are characterized by varying $SiO_2$ (45.7-55.5 wt.%)
and low Mg# values (45-52), suggesting that they were not derived from the
primary magmas, and that they likely experienced crustal assimilation and
fractional crystallization (AFC) processes. Generally, mantle - derived magmas
suffer various degrees of crust contamination en-route from magma chambers
to the surface (Aitcheson and Forrest, 1994). The presence of inherited
Paleozoic and Neoproterozoic zircons in these basalts suggests the potential



interactions between the ascending magmas and the country rocks (Fig. 5a).
However, these basaltic rocks exhibit no negative anomalies of Nb, Ta, and Ti,
which are typically depleted in the crust (Fig. 7d). They exhibit low La/Nb ratios
(0.53 - 1.15) and mostly have high Nb/U ratios (37 - 45), similar to the range of
oceanic lavas (La/Nb <1.2 and Nb/U >39; Krienitz et al., 2006). Additionally, all
basalt samples exhibit low Th/Nb ratios (0.09-0.15), plotting along the
MORB−OIB array of oceanic basalts within the Th/Yb-Nb/Yb diagram (Fig. 7e;
Pearce, 2008). These signatures, with little indication of crustal components,
suggest that these basalts experienced negligible contamination during their
journey to the surface. They are characterized by extremely low concentrations
of Ni (27.4–61.2 ppm) and Cr (25.4–108 ppm). They also exhibit slight negative
anomalies of Eu and Sr on the whole-rock normalized REE patterns and spider
diagram (Fig. 7c and 7d). These features could be caused by varying degrees
of fractional crystallization processes involving olivine, clinopyroxene, and
plagioclase.
The Early Jurassic episode of volcanism in the West Kunlun Mountains
temporally followed the Cimmerian Orogeny. Regionally, the eruption of basalts
at 178 Ma was slightly later than the peak metamorphism of high-pressure
granulite facies that has been proposed to have occurred between 200 and 185
Ma (Qu et al., 2021). Collisional orogeny commonly transitions from syn-
collisional metamorphism to post-collisional unroofing (Dilek and Altunkaynak,
2007, 2010; Zheng et al., 2019). The unroofing phase could generate
geochemically varying granitoids with extrusion of mafic magma (Harris et al.,
1986; Zhou et al., 2021). However, distinguishing post-collisional from syn-
collisional magmatism may present challenges, because the post-collisional



mafic rocks could inherit whole-rock geochemical fingerprints from the
preceding subducted materials (Zhao et al., 2013). Conversely, intraplate
magmas are typically dominated by low-degree partial melting and silica-
unsaturated alkaline magmas, which is distinct from syn- and post-collisional
igneous rocks (Dilek and Altunkaynak, 2010; Xu et al., 2020).
The Jurassic alkali basalts exhibit enrichment of LREE and HSFEs without
obvious crustal signatures (e.g., Nb-Ta depletion; Fig. 7c-d), different from the
syn- and post-collisional magmas in the West Kunlun Mountains (Liao et al.,
2012; Chen et al., 2021). Their compositions resemble those of intraplate OIBs
and could have been generated by low-degree partial melting (~5%) of a garnet
lherzolite mantle source (Fig. 7e-f). All tectonic discrimination plots using
immobile trace elements indicate that the Jurassic basalts formed within an
intraplate setting (Fig. 8).
The generation of these magmas can be attributed to one of two
mechanisms. The first explanation is that the North Kunlun region experienced
rapid orogenic collapse after Late Triassic collisional orogeny, during which
intra-plate collapse-related volcanism generate the observed basalt flows. We
do not find this hypothesis plausible given the implied rapid transition from peak
collisional orogeny, including ca. 185 Ma prograde metamorphism, to collapse
and volcanism recorded at ca. 175 Ma (Wu et al., 2021). Many arc-continent or
continent-continent collisional orogens, evolving from peak orogenic
metamorphism, to orogenic collapse, to intraplate stage, collectively persist for
tens of millions of years (Dewey, 2005; Weller et al., 2021).
Conversely, a broad plate-boundary extensional process may have
impacted this orogenic belt and its hinterland region in the Early Jurassic.



Support for this model includes the expansive extensional rifts developed
across the interior Eurasia during the Early-Middle Jurassic (e.g., Amu–Dar'ya,
Afghan–Tajik and Fergana basins; Otto, 1997). The opening of the Greater
Caucasus - proto-South Caspian Sea back-arc basin at the southern Eurasian
margin nearly at the same time has been ascribed to a slab retreat event within
the Neo-Tethys (Golonka, 2004). Back-arc transgression and MORB-liked
magmas have been also identified in the Tianshuihai terrane (Fig. 7; Jian et al.,
2019), suggesting the slab-pull effect on the studied region in the West Kunlun
Mountains. In this scenario, the Early Jurassic basalts were generated during
regional extension across the region, accompanied by intra-plate volcanism.

**6.2 Jurassic basin formation and implications for sedimentary**
**provenance**
The closure of the Paleo-Tethyan Ocean led to collision of the Cimmerian
terranes with Eurasia that caused the development of a regional unconformity
across the central Asia during the Triassic to Early Jurassic (Gaetani et al., 1993;
Schwab et al., 2004; Fürsich et al., 2017). This orogenic unconformity
separates the imbricated Triassic flysch strata below from the overlying Middle
Jurassic limestones in the Tianshuihai-Qiangtang block (Zhao et al., 2000). In
the studied area, the deformed Upper Paleozoic strata are unconformably
overlain by a Lower Jurassic conglomerate (Fig. 9). Analysis of the Lower
Jurassic deposits suggests a regional transtension following the Cimmerian
collision (Sobel, 1999). Analysis of the available seismic data identifies the
Jurassic horst-graben patterns, favoring the extensional setting within basin
interior (Zhao et al., 2020; Li et al., 2022a).





The Kyzyltau basin preserves the most comprehensive record of the

formation and evolution of a post-Cimmerian rift, spanning from its initiation in
the Early Jurassic to its inversion in the Late Jurassic (Wu et al., 2021). The
basement of this basin varies along its lateral extent, indicating its strong
tectonic reworking prior to Jurassic deposition. It comprises four subdivisions
from the north to the southeast: (1) An Early Devonian metasedimentary rock
terrane in the Kashgar depression (1-4 in Fig. 9), (2) The Carboniferous island-
arc crust and Permian back-arc basin successions in the northwestern segment
of the West Kunlun orogenic belt (5-6 in Fig. 9), (3) An Upper Carboniferous to
Middle Permian platform successions in the middle segment (7-11 in Fig. 9),
and (4) An Upper Permian clastic formation in the southern part (12-17 in Fig.

9).

The massive conglomerate of the Shalitashi Formation indicates rapid

infilling of the Jurassic basin during its initial opening stage in the West Kunlun
orogenic belt. Analysis of conglomerate clast lithologies suggests that different
sites exhibit sharp variations in their compositions, consistent with the presence
of local basement rocks (Fig. 9). For example, the gravels in the Kashgar
depression are mainly derived from sandstone strata, pointing to the source of
the underlying Devonian (Wulagen) uplift. The gravels from the Oytag and Gaizi
sections show complex compositions, with abundant igneous and siliceous rock
fragments, which might have been provided by the local arc and back-arc basin
lithologies. Contrastively, gravels from the Tamu section are composed
predominately of limestones, implying their origin from the underlying
Carboniferous marine strata. Gravels from the Qimugen and Kusilafu sections
share a similar arenaceous source region, which exists in the Devonian and



Permian strata in the core of the Kashgar-Yecheng syncline (Fig. 2a).

The Lower Jurassic strata rapidly transition from alluvial fan deposits into

fluvial sedimentary environment, which is indicated by the Middle Jurassic,
stacked coal-bearing sandstones of the Kangsu Formation (Fig. 9). During the
Middle Jurassic, extensional faulting across the half-grabens further deepened
the basin and facilitated the deposition of a turbidite sequence of the Yangye
Formation (Wu et al., 2021). Provenance analysis based on detrital zircon age
dating suggests that the source region for these sandstones was dominated by
Late Ordovician-Early Silurian (~ 446 - 435 Ma) and Neoproterozoic (~ 980 -
780 Ma) igneous rocks, with minor Neoarchean-Paleoproterozoic and
Mesoproterozoic ages (Fig. 6). Early Paleozoic (~ 480 - 440 Ma) granitoids,
with a peak intrusive at ~ 440 Ma (Fig. 1c; Tao et al., 2024), are exposed
extensively in the South Kunlun terrane. However, the South Kunlun terrane is
unlikely to be the source for these Jurassic depositions because the South
Kunlun region contains extensive Triassic (~ 240 - 210 Ma) granitoids, intruded
into the early Paleozoic rock units (Fig. 1c; Chen et al., 2021). Yet, Triassic
detrital zircons are absent in the Lower - Middle Jurassic strata (Fig. 6).
Therefore, we instead suggest that the potential source area was most likely
the North Kunlun terrane, which consists mainly of Paleozoic strata and
Precambrian metamorphic basement lithologies. A provenance study has
revealed that the age patterns of detrital zircons from the Ordovician - Devonian
strata contain main age peaks at 430 - 445 Ma, 930 - 800 Ma, and 790 - 760
Ma, with subordinate Neoarchean to Mesoproterozoic ages (Yan, 2022). Our
results are consistent with this detrital zircon age information from the Lower
Paleozoic sedimentary rocks and with the paleocurrent results of previous





studies (Wu et al., 2021). The findings from detrital zircon analyses are also
compatible with the constraints from clast lithologies in the Lower Jurassic
conglomerate, indicating a proximal feature of the source- to-sink system
developed in the half grabens.
A Late Jurassic contractional event affected this region, as evidenced by
the intense deformation and metamorphism displayed by various formations
and rock units (Robinson et al., 2007; Groppo et al., 2019), and by the uplift
and inversion of the earlier basin (Yang et al., 2017). The Middle Jurassic
shallow marine sequences in Qiangtang and Pamir were uniformly eroded
during this time period. The Upper Jurassic strata are either entirely absent or
locally replaced by conglomerate deposits (Fig. 10). In the southern Tarim Basin,
the Upper Jurassic strata are dominated by brownish reddish conglomerate of
the Kuzigongsu Formation. Previous studies have suggested that these
redbeds may have signalled a regional increase in aridity and the cessation of
the monsoons as a result of the uplift of the surrounding mountain belts
(Hendrix, 2000). A Late Jurassic uplift event, which significantly impacted the
basinal tectonostratigraphy, has been corroborated by numerous
thermochronologic ages (170-155 Ma) within the West Kunlun Mountains and
Pamir (Fig.1c; Yang et al., 2017). The inferred uplift event also resulted in
significant changes in basin and range patterns, and influenced the potential
provenance of sediments. The emergence of juvenile detrital zircons in these
Upper Jurassic and Lower Cretaceous deposits indicates the exhumation and
erosion of a late Paleozoic to Mesozoic arc system (Fig. 10). The Triassic
batholiths were thrust onto the southwestern margin of the Tarim Basin creating
an elevated topography, which in turn provided abundant clastic material into





the Cretaceous depocenters in the region.

**6.3 Switching extensional and contractional tectonics related to the**

**subduction of Neo-Tethys**

The Mesozoic era records the transition from the closure of the Paleo-

Tethys Ocean to the initiation of subduction within Neo-Tethys (Wan et al., 2019).
These processes are influenced by complex plate tectonic conditions, as the
evolution of the Paleo- and Neo-Tethys Oceans varies significantly in their time-
space patterns. The two Tethyan seaways diverge into several branches
extending from Iran to Pamir, then eastward into the Tibetan Plateau (Fig. 1a).
Deciphering the history of the Pamir Tethyan segment, therefore, improves our
knowledge of the geodynamic evolution of the entire Tethyan realm.

Two major tectonic events profoundly affected the sedimentary patterns of

the Mesozoic successions in this region. Episodic collisions along the southern
Asian margin in the Late Triassic and then in the Late Jurassic resulted in major
deformation (Jolivet, 2017). The regional magmatic history and the results of
the provenance studies of the Jurassic basin necessitate a geodynamic
scenario to explain the mechanism of an extensional tectonic event between
two major contractional events. Although a flat subduction model has recently
been proposed to explain the regional Cretaceous magmatism in the Pamir, the
mode of Jurassic tectonic processes remains poorly constrained (Chapman et
al., 2018). As discussed above, the history of the Neo-Tethyan subduction
events significantly varies spatially. The initiation of subduction along the
Tibetan margin occurred during the Middle Triassic, leading to volcanic
activities in the southern Lhasa (Wang et al., 2016; Xie et al., 2021), whereas



the subduction in the Iran sector in the same orogenic belt farther west initiated
later in the Early Jurassic (Wan et al., 2023). The extensive Early-Middle arc
Jurassic magmatism along both continental margins indicates a synchronous
flare-up of continental arcs (Fig. 11a and 11c). The bimodal volcanism (195-174
Ma) in the Gangdese arc was associated with the subsequent opening of a
back-arc basin (174-156 Ma) (Fig. 11c; Kapp and DeCelles, 2019). The
magmatic arc of the Sanandaj–Sirjan belt (180-140 Ma) in SW Iran was
facilitated by a simultaneous progressive back-arc rift (Fig. 11a; Hassanzadeh
and Wernicke, 2016; Azizi and Stern, 2019).

By comparison, compiled magmatic detrital zircons in the Pamir segment

reveal that Early-Middle Jurassic magmatism was almost absent there (Fig. 11b;
Chapman et al., 2018). Available geochronological data indicate that Jurassic
igneous rocks surrounding the Pamir are also limited, with only basalts exposed
in the North Kunlun (Kandilik) and Tianshuihai regions (Jian et al., 2019) and
bimodal volcanic rock suites found in the east of Karakoram (Zhou et al., 2019).
Geochemical studies reveal that these coeval basaltic lavas (178-174 Ma)
exhibit distinct features in their major and trace element compositions (Fig. 7
and 8). Magmas of the basaltic lavas in the North Kunlun were dominated by
within-plate basalts that shared similar compositions with typical OIB. In
contrast, basalts in the Tianshuihai to the south were dominated by back-arc
MORBs, characterized by distinct Nb-Ta depletions. The scarcity of zircon-rich
felsic magmas in this region evidently differs from the conditions in the western
and eastern segments of the Eurasian Tethyan margins where arc magmatism
developed upon continental basement. To date, the exact timing of the onset of
subduction-related magmatism in the Pamir Tethyan margin remains unclear.





The geochronological dataset for the Karakoram arc and the Kohistan Ladakh
arc indicates that magmatic activity may have occurred as early as the Late
Jurassic (Fig. 11b; Jagoutz et al., 2018; Saktura et al., 2023).
While the spatial continuity of the Tethyan suture zones from Iran into Tibet
remains enigmatic, we propose that the regional Early to Middle Jurassic
extension expressed across the southern Eurasian continental margin was a
consequence of retreating subduction of the Neo-Tethyan Ocean floor. First,
the transition from Cimmerian orogenic build-up (200-185 Ma) to large-scale
continental extension (178-174 Ma) suggests the involvement of additional
external extensional stresses, different from the classic cases of continent -
continent collision (Weller et al., 2021). No typical post-collisional mafic igneous
rock has been identified in the West Kunlun Orogenic Belt as of now. Secondly,
the 195 Ma bimodal volcanic rocks in Karakoram and the 174 Ma MORB-like
basalts in Tianshuihai have been suggested as associated with the initial
opening of a back-arc basin, based on their geochemical signatures of crustal
material metasomatism (Jian et al., 2019; Zhou et al., 2019). The magmatism
in Pamir and Karakoram was quite similar to the extensional episodes that
occurred in the southern margin of the Lhasa block, caused by accelerated slab
rollback (Kapp and DeCelles, 2019). Thirdly, deposition of shallow marine
carbonates was prevalent in the Pamir and Karakoram during the Middle
Jurassic (Fig. 10), indicating an expansive extensional continental platform
facing the ocean (Yang et al., 2017). These scenarios are analogous to the
active margin of the western Pacific rim, which is characterized by a broad
marginal sea with an outboard trench - subduction chain (Fig. 1a). Additionally,
the Middle Jurassic extension occurred across the broad hinterlands of central



Asia, which cannot be easily explained by the collapse of the Paleo-Tethyan
orogenic belt (Otto, 1997).
During the Late Jurassic, this marginal extensional basin started to invert,
with extensive contractional deformation of the Lower-Middle Jurassic
carbonate strata and the development of a major angular unconformity (Gaetani
et al., 1993; Robinson, 2015). Available basement thermochronological data
show widespread exhumation across the West Kunlun Mountains (Fig. 1c), as
well as the reactivation of the Paleo-Tethyan sutures within the Pamir terranes
(Schwab et al., 2004). The exhumation of the Triassic plutons in the South
Kunlun Mountain led to the transport of debris material from the magmatic arc
into the Tarim basin through braided fluvial network systems (Fig. 11b). This
broad uplift event has been interpreted as retro-arc deformation and shortening
related to the advancing subduction of the Neo-Tethyan Ocean (Robinson,

2015).

The subduction style along the broader strike-length of the Tethyan orogen
varied from the west to the east in the Late Jurassic - Early Cretaceous. Similar
to the West Kunlun Mountains, the Lhasa block to the east experienced basin
inversion and contractional deformation starting by ca. 155 Ma and throughout
the Early Cretaceous (e.g., Murphy et al., 1997; Ding and Lai, 2003; Kapp and
DeCelles, 2019). Geological mapping has documented significant shortening
strain (~ 60%) across Lhasa at this time (Murphy et al., 1997). Although the
cause of this event has been debated, the magmatic lull since the earliest
Cretaceous and subsequent flare-up in the Mid-Cretaceous in both regions
imply that they shared a similar geodynamic setting (Fig. 11; Chapman et al.,
2018). Conversely, the Iranian segment to the west experienced continuous





extension at the same time (Hunziker et al., 2015; Lechmann et al., 2018;
Maghdour-Mashhour et al., 2021). These along-strike variations likely reflect
broad geodynamic changes to, or initial conditions of, the Tethyan Ocean
system that warrant future investigations. For example, variable plate
convergence rates related to global tectonic configurations or the oceanic-plate
age variations could result in unique tectonic events along the strike-length of
the entire Tethyan orogen. Alternatively, the closure of the Bangong-Nujiang
Ocean, another branch of the Tethyan system between the Lhasa and
Qiangtang blocks, might have also played a significant role in along-strike
variations within the Tethyan orogenic belt (Fig. 11; Yang et al., 2017; Kapp and
DeCelles, 2019).

**7 Conclusion**
This study has concentrated on the stratigraphy and provenance of
Jurassic strata in the West Kunlun Mountains to better understand the
Mesozoic geological evolution of the Eurasian margin within the framework of
the Tethyan geodynamics. Our investigations of the Jurassic sedimentary
successions, combined with new geochronological and geochemical data from
coeval basaltic lava intercalations, led to the following conclusions:
(1) A newly identified, thick sedimentary package with basaltic lava
interlayers in the southern end of the Kyzyltau basin bears similarities to the
Lower and Middle Jurassic sequences in their clastic compositions and
structures. Zircon U-Pb dating results from basaltic lavas suggest an Early
Jurassic age (~ 178 Ma) for this stratigraphic member, in contrast to a
Precambrian age previously reported. This is a significant change that strongly



affects the current tectonic interpretations and models.
(2) Our new geochemical data from the Early Jurassic basaltic extrusive
rocks show that magmas of these basalts had typical OIB affinities, and that
they lacked crustal contamination. Thus, the related magmatism likely occurred
in an intraplate rifting setting and was facilitated by extensional fault systems,
which significantly reduced the residence time of the ascending magmas in the
crust avoiding contamination.
(3) Provenance analysis, integrating conglomerate clast lithologies with
detrital zircons, indicates a significant source contribution from local basements
(North Kunlun) for the Early to Middle Jurassic rift basins. In comparison, the
Late Jurassic contractional event caused an uplift of the surrounding mountains
in the South Kunlun and Pamir, significantly influencing the basin
tectonostratigraphy and source- to -sink system.
(4) The alternating extensional and contractional tectonic episodes in the
West Kunlun Mountains and a wider region across the southern Eurasian
margin are related to changes in the subduction style of the Neo-Tethyan Ocean
floor, transitioning from retreating in Early - Middle Jurassic to advancing in Late
Jurassic - Early Cretaceous.

**Declaration of Competing Interest**
The authors declare that they have no known competing financial interests
or personal relationships that could have appeared to influence the work
reported in this paper.



**Acknowledgement**

This work was supported by the National Natural Science Foundation of China (Grants No. U22B6002 and 42302231). H.-X. Wu received the funding of Postdoctoral Science Foundation (2023M742979).

**Author Contributions**

*Hong-Xiang Wu:* Conceptualization, Formal Analysis, Investigation, Methodology, Visualization, Writing – original draft, Writing – review & editing, Funding acquisition; *Han-Lin Chen:* Funding acquisition, Investigation, Project administration; *Andrew V. Zuza: Writing – review & editing; Yildirim Dilek: Writing – review & editing; Du-Wei Qiu:* Investigation, Formal Analysis; *Qi-Ye Lu:* Investigation, Formal Analysis; *Feng-Qi Zhang:* Investigation, Formal Analysis; *Xiao-Gan Cheng:* Investigation*; Xiu-Bin Lin:* Investigation.

**Data availability**

The data used in this study are available in the references and Supplementary Material, including five tables and one figure.



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

998        M.: From Jurassic rifting to Cretaceous subduction in NW Iranian



Azerbaijan: geochronological and geochemical signals from granitoids,
Contributions to Mineralogy and Petrology, 173, 102, 10.1007/s00410-018-

1532-8, 2018.

Leith, W.: A mid-Mesozoic extension across Central Asia?, Nature, 313, 567-
570, 10.1038/313567a0, 1985.
Li, G., Sandiford, M., Fang, A., Kohn, B., Sandiford, D., Fu, B., Zhang, T., Cao,
Y., and Chen, F.: Multi-stage exhumation history of the West Kunlun orogen
and the amalgamation of the Tibetan Plateau, Earth and Planetary Science
Letters, 528, 115833, https://doi.org/10.1016/j.epsl.2019.115833, 2019.
Li, L., Najman, Y., Dupont-Nivet, G., Parra, M., Roperch, P., Kaya, M., Meijer,
N., O'Sullivan, P., Jepson, G., and Aminov, J.: Mesozoic–Cenozoic
multistage tectonic evolution of the Pamir: Detrital fission-track constraints
from the Tajik Basin, Basin Research, 35, 530-550,
https://doi.org/10.1111/bre.12721, 2023.
Li, S., Zhao, S., Liu, X., Cao, H., Yu, S., Li, X., Somerville, I., Yu, S., and Suo,
Y.: Closure of the Proto-Tethys Ocean and Early Paleozoic amalgamation
of microcontinental blocks in East Asia, Earth-Science Reviews, 186, 37-
75, https://doi.org/10.1016/j.earscirev.2017.01.011, 2018.
Li, Y., Robinson, A. C., Zucali, M., Gadoev, M., Oimuhammadzoda, I., Lapen, T.
J., and Carrapa, B.: Mesozoic Tectonic Evolution in the Kurgovat-Vanch
Complex, NW Pamir, Tectonics, 41, e2021TC007180,
https://doi.org/10.1029/2021TC007180, 2022b.
Li, Y., Wen, L., Yang, X.-Z., Li, C., Zhang, L., Wang, B., Chen, C., Liu, Y.-L., and
Li, Y.-J.: Mesozoic Collision-Related Structures in the Southern Tarim
Basin, W. China: Implications for the Paleo-Tethys Closing Process,



Frontiers in Earth Science, 9, 10.3389/feart.2021.792049, 2022a.

Liao, S., Jiang, Y., Zhou, Q., Yang, W., Jin, G., and Zhao, P.: Geochemistry and geodynamic implications of the Triassic bimodal magmatism from Western Kunlun Orogen, northwest China, International Journal of Earth Sciences, 101, 555-577, 10.1007/s00531-011-0686-7, 2012.

Ma, S., Wang, Y., and Fang, X.: Basic characteristics of Proterozoic Eonothem as a table cover on northern slope, Xinjiang Geology, 9, 59-71, 1991 (in Chinese with English abstract).

Ma, X., Xu, Z., Meert, J., and Santosh, M.: Early Jurassic intra-oceanic arc system of the Neotethys Ocean: Constraints from andesites in the Gangdese magmatic belt, south Tibet, Island Arc, 26, e12202, https://doi.org/10.1111/iar.12202, 2017.

Maghdour-Mashhour, R., Hayes, B., Pang, K.-N., Bolhar, R., Tabbakh Shabani, A. A., and Elahi-Janatmakan, F.: Episodic subduction initiation triggered Jurassic magmatism in the Sanandaj–Sirjan zone, Iran, Lithos, 396-397, 106189, https://doi.org/10.1016/j.lithos.2021.106189, 2021.

Mattern, F. and Schneider, W.: Suturing of the Proto- and Paleo-Tethys oceans in the western Kunlun (Xinjiang, China), Journal of Asian Earth Sciences, 18, 637-650, https://doi.org/10.1016/S1367-9120(00)00011-0, 2000.

Meschede, M.: A method of discriminating between different types of mid-ocean ridge basalts and continental tholeiites with the Nb  1bZr  1bY diagram, Chemical Geology, 56, 207-218, https://doi.org/10.1016/0009-2541(86)90004-5, 1986.

Metcalfe, I.: Gondwana dispersion and Asian accretion: Tectonic and palaeogeographic evolution of eastern Tethys, Journal of Asian Earth



Sciences, 66, 1-33, 10.1016/j.jseaes.2012.12.020, 2013.
Metcalfe, I.: Multiple Tethyan ocean basins and orogenic belts in Asia,
Gondwana Research, 100, 87-130,
https://doi.org/10.1016/j.gr.2021.01.012, 2021.
Middlemost, E. A. K.: Naming materials in the magma/igneous rock system,
Earth-Science Reviews, 37, 215-224, https://doi.org/10.1016/0012-

8252(94)90029-9, 1994.

Murphy, M. A., Yin, A., Harrison, T. M., Dürr, S. B., Z, C., Ryerson, F. J., Kidd,
W. S. F., X, W., and X, Z.: Did the Indo-Asian collision alone create the
Tibetan plateau?, Geology, 25, 719-722, 10.1130/0091-
7613(1997)025<0719:Dtiaca>2.3.Co;2, 1997.
Otto, S. C.: Mesozoic-Cenozoic history of deformation and petroleum systems
in sedimentary basins of Central Asia; implications of collisions on the
Eurasian margin, Petroleum Geoscience, 3, 327-341,
10.1144/petgeo.3.4.327, 1997.
Pearce, J. A.: Geochemical fingerprinting of oceanic basalts with applications
to ophiolite classification and the search for Archean oceanic crust, Lithos,
100, 14-48, https://doi.org/10.1016/j.lithos.2007.06.016, 2008.
Pearce, J. A.: Trace element characteristics of lavas from destructive plate
boundaries, in: Orogenic Andesites and Related Rocks, edited by: Thorpe,
R. S., John Wiley and Sons, Chichester, England, 525-548, 1982.
Qu, J., Zhang, L., Zhang, J., and Zhang, B.: Petrology and geochronology on
high-pressure pelitic granulite from Bulunkuole complex in West Kunlun
and its tectonic implication, Acta Petrologica Siniaca, 37, 563-574,

10.18654/1000-0569/2021.02.14, 2021.



Robinson, A. C.: Mesozoic tectonics of the Gondwanan terranes of the Pamir
plateau, Journal of Asian Earth Sciences, 102, 170-179,
https://doi.org/10.1016/j.jseaes.2014.09.012, 2015.
Robinson, A. C., Yin, A., Manning, C. E., Harrison, T. M., Zhang, S.-H., and
Wang, X.-F.: Cenozoic evolution of the eastern Pamir: Implications for
strain-accommodation mechanisms at the western end of the Himalayan-
Tibetan orogen, GSA Bulletin, 119, 882-896, 10.1130/b25981.1, 2007.
Rollinson, H. R.: Using Geochemical Data: Evaluation, Presentation,
Interpretation, Mineralogical Magazine, Longman, Edinburgh Gate,
London, 352 pp.1993.
Ruban, D. A., Al-Husseini, M. I., and Iwasaki, Y.: Review of Middle East
Paleozoic plate tectonics, GeoArabia, 12, 35-56,
10.2113/geoarabia120335, 2007.
Saktura, W. M., Buckman, S., Nutman, A. P., Walsh, J., and Murray, G.:
Magmatic records from the Karakoram terrane: U–Pb zircon ages from
granites and modern sediments in the Nubra Valley, NW Himalaya, Journal
of Asian Earth Sciences, 255, 105771,
https://doi.org/10.1016/j.jseaes.2023.105771, 2023.
Schwab, M., Ratschbacher, L., Siebel, W., McWilliams, M., Minaev, V., Lutkov,
1093        V., Chen, F., Stanek, K., Nelson, B., Frisch, W., and Wooden, J. L.:
Assembly of the Pamirs: Age and origin of magmatic belts from the
southern Tien Shan to the southern Pamirs and their relation to Tibet,
Tectonics, 23, https://doi.org/10.1029/2003TC001583, 2004.
Şengör, A. M. C.: Mid-Mesozoic closure of Permo–Triassic Tethys and its
implications, Nature, 279, 590-593, 10.1038/279590a0, 1979.



Şengör, A. M. C.: The Cimmeride Orogenic System and the Tectonics of Eurasia, in: The Cimmeride Orogenic System and the Tectonics of Eurasia, Geological Society of America, 0, 10.1130/SPE195-p1, 1984.

Şengör, A. M. C.: Tectonics of the Tethysides: Orogenic Collage Development in a Collisional Setting, Annual Review of Earth and Planetary Sciences, 15, 213-244, https://doi.org/10.1146/annurev.ea.15.050187.001241, 1987.

Şengör, A. M. C., Altıner, D., Cin, A., Ustaömer, T., and Hsü, K. J.: Origin and assembly of the Tethyside orogenic collage at the expense of Gondwana Land, Geological Society, London, Special Publications, 37, 119-181, doi:10.1144/GSL.SP.1988.037.01.09, 1988.

Sobel, E. R.: Basin analysis of the Jurassic–Lower Cretaceous southwest Tarim basin, northwest China, GSA Bulletin, 111, 709-724, 10.1130/0016-7606(1999)111<0709:Baotjl>2.3.Co;2, 1999.

Sobel, E. R., Chen, J., Schoenbohm, L. M., Thiede, R., Stockli, D. F., Sudo, M., and Strecker, M. R.: Oceanic-style subduction controls late Cenozoic deformation of the Northern Pamir orogen, Earth and Planetary Science Letters, 363, 204-218, https://doi.org/10.1016/j.epsl.2012.12.009, 2013.

Stampfli, G., Marcoux, J., and Baud, A.: Tethyan margins in space and time, Palaeogeography, Palaeoclimatology, Palaeoecology, 87, 373-409, https://doi.org/10.1016/0031-0182(91)90142-E, 1991.

Stampfli, G. M.: Tethyan oceans, Geological Society, London, Special Publications, 173, 1-23, doi:10.1144/GSL.SP.2000.173.01.01, 2000.

Stampfli, G. M. and Borel, G. D.: A plate tectonic model for the Paleozoic and Mesozoic constrained by dynamic plate boundaries and restored synthetic oceanic isochrons, Earth and Planetary Science Letters, 196, 17-33,





https://doi.org/10.1016/S0012-821X(01)00588-X, 2002.
Sun, S.-S. and McDonough, W. F.: Chemical and isotopic systematics of
oceanic basalts: implications for mantle composition and processes,
Geological Society, London, Special Publications, 42, 313-345,
doi:10.1144/GSL.SP.1989.042.01.19, 1989.

Tapponnier, P., Mattauer, M., Proust, F., and Cassaigneau, C.: Mesozoic
ophiolites, sutures, and arge-scale tectonic movements in Afghanistan,
Earth    and    Planetary    Science    Letters,    52,    355-371,
https://doi.org/10.1016/0012-821X(81)90189-8, 1981.

Tao, Z., Yin, J., Spencer, C. J., Sun, M., Xiao, W., Kerr, A. C., Wang, T., Huangfu,
P., Zeng, Y., and Chen, W.: Subduction polarity reversal facilitated by plate
coupling during arc-continent collision: Evidence from the Western Kunlun
orogenic belt, northwest Tibetan Plateau, Geology, 10.1130/g51847.1,
2024.

Vermeesch, P.: IsoplotR: A free and open toolbox for geochronology,
Geoscience    Frontiers,    9,    1479-1493,
https://doi.org/10.1016/j.gsf.2018.04.001, 2018.

Vermeesch, P.: On the visualisation of detrital age distributions, Chemical
Geology,    312-313,    190-194,
https://doi.org/10.1016/j.chemgeo.2012.04.021, 2012.

Wan, B., Chu, Y., Chen, L., Zhang, Z., Ao, S., and Talebian, M.: When and Why
the NeoTethyan Subduction Initiated Along the Eurasian Margin, in:
Compressional    Tectonics,    245-260,
https://doi.org/10.1002/9781119773856.ch9, 2023.

Wan, B., Wu, F., Chen, L., Zhao, L., Liang, X., Xiao, W., and Zhu, R.: Cyclical



one-way continental rupture-drift in the Tethyan evolution: Subduction-
driven plate tectonics, Science China Earth Sciences, 62, 2005-2016,
10.1007/s11430-019-9393-4, 2019.
Wang, C., Ding, L., Zhang, L.-Y., Kapp, P., Pullen, A., and Yue, Y.-H.:
Petrogenesis of Middle–Late Triassic volcanic rocks from the Gangdese
belt, southern Lhasa terrane: Implications for early subduction of Neo-
Tethyan oceanic lithosphere, Lithos, 262, 320-333,
https://doi.org/10.1016/j.lithos.2016.07.021, 2016.
Wang, Y., Qian, X., Cawood, P. A., Liu, H., Feng, Q., Zhao, G., Zhang, Y., He,
H., and Zhang, P.: Closure of the East Paleotethyan Ocean and
amalgamation of the Eastern Cimmerian and Southeast Asia continental
fragments, Earth-Science Reviews, 186, 195-230,
https://doi.org/10.1016/j.earscirev.2017.09.013, 2018.
Wei, Y., Zhao, Z., Niu, Y., Zhu, D.-C., Liu, D., Wang, Q., Hou, Z., Mo, X., and
Wei, J.: Geochronology and geochemistry of the Early Jurassic Yeba
Formation volcanic rocks in southern Tibet: Initiation of back-arc rifting and
crustal accretion in the southern Lhasa Terrane, Lithos, 278-281, 477-490,
https://doi.org/10.1016/j.lithos.2017.02.013, 2017.
Weller, O. M., Mottram, C. M., St-Onge, M. R., Möller, C., Strachan, R., Rivers,
T., and Copley, A.: The metamorphic and magmatic record of collisional
orogens, Nature Reviews Earth & Environment, 2, 781-799,
10.1038/s43017-021-00218-z, 2021.
Winchester, J. A. and Floyd, P. A.: Geochemical discrimination of different
magma series and their differentiation products using immobile elements,
Chemical Geology, 20, 325-343, https://doi.org/10.1016/0009-



1174 2541(77)90057-2, 1977.

1175 Wu, C., Yin, A., Zuza, A. V., Zhang, J., Liu, W., and Ding, L.: Pre-Cenozoic

1176 geologic history of the central and northern Tibetan Plateau and the role of

1177 Wilson cycles in constructing the Tethyan orogenic system, Lithosphere, 8,

1178 254-292, 10.1130/l494.1, 2016.

1179 Wu, F. Y., Wan, B., Zhao, L., Xiao, W. J., and Zhu, R. X.: Tethyan geodynamics,

1180 Acta Petrologica Siniaca, 36, 1627-1674, 2020.

1181 Wu, H., Cheng, X., Chen, H., Chen, C., Dilek, Y., Shi, J., Zeng, C., Li, C., Zhang,

1182 W., Zhang, Y., Lin, X., and Zhang, F.: Tectonic Switch From Triassic

1183 Contraction to Jurassic-Cretaceous Extension in the Western Tarim Basin,

1184 Northwest China: New Insights Into the Evolution of the Paleo-Tethyan

1185 Orogenic Belt, Frontiers in Earth Science, 9, 10.3389/feart.2021.636383,

1186 2021.

1187 Xiao, W. J., Windley, B. F., Chen, H. L., Zhang, G. C., and Li, J. L.:

1188 Carboniferous-Triassic subduction and accretion in the western Kunlun,

1189 China: Implications for the collisional and accretionary tectonics of the

1190 northern Tibetan Plateau, Geology, 30, 295-298, 10.1130/0091-

1191 7613(2002)030<0295:Ctsaai>2.0.Co;2, 2002.

1192 Xiao, W. J., Windley, B. F., Liu, D. Y., Jian, P., Liu, C. Z., Yuan, C., and Sun, M.:

1193 Accretionary Tectonics of the Western Kunlun Orogen, China: A Paleozoic–

1194 Early Mesozoic, Long‐Lived Active Continental Margin with Implications

1195 for the Growth of Southern Eurasia, The Journal of Geology, 113, 687-705,

1196 10.1086/449326, 2005.

1197 Xie, F. and Tang, J.: The Late Triassic-Jurassic magmatic belt and its

1198 implications for the double subduction of the Neo-Tethys Ocean in the



southern Lhasa subterrane, Tibet, Gondwana Research, 97, 1-21,
https://doi.org/10.1016/j.gr.2021.05.007, 2021.
Xie, Y. and Dilek, Y.: Detrital zircon U–Pb geochronology and fluvial basin
evolution of the Liuqu Conglomerate within the Yarlung Zangbo Suture
Zone: A critical geochronometer for the collision tectonics of the Tibetan-
Himalayan Orogenic Belt, Geosystems and Geoenvironment, 2, 100178,
https://doi.org/10.1016/j.geogeo.2023.100178, 2023.
Xu, W., Zhao, Z., and Dai, L.: Post-collisional mafic magmatism: Record of
lithospheric mantle evolution in continental orogenic belt, Science China
Earth Sciences, 63, 2029-2041, 10.1007/s11430-019-9611-9, 2020.
Yan, J.: The early Paleozoic tectono-sedimentary characteristics and the basin-
orogen process in south Tarim Basin, School of Earth Sciences, Zhejiang
University, Hangzhou, Zhejiang, China, 137 pp.,
10.27461/d.cnki.gzjdx.2022.002783, 2022 (in Chinese with English
abstract).
Yang, Y.-T., Guo, Z.-X., and Luo, Y.-J.: Middle-Late Jurassic
tectonostratigraphic evolution of Central Asia, implications for the collision
of the Karakoram-Lhasa Block with Asia, Earth-Science Reviews, 166, 83-
110, https://doi.org/10.1016/j.earscirev.2017.01.005, 2017.
Zhang, K.-J., Zhang, Y.-X., Tang, X.-C., and Xia, B.: Late Mesozoic tectonic
evolution and growth of the Tibetan plateau prior to the Indo-Asian collision,
Earth-Science Reviews, 114, 236-249,
https://doi.org/10.1016/j.earscirev.2012.06.001, 2012.
Zhang, Q., Wu, Z., Chen, X., Zhou, Q., and Shen, N.: Proto-Tethys oceanic slab
break-off: Insights from early Paleozoic magmatic diversity in the West



Kunlun Orogen, NW Tibetan Plateau, Lithos, 346-347, 105147,
https://doi.org/10.1016/j.lithos.2019.07.014, 2019a.
Zhang, S., Hu, X., and Garzanti, E.: Paleocene initial indentation and early
growth of the Pamir as recorded in the western Tarim Basin,
Tectonophysics, 772, 228207, https://doi.org/10.1016/j.tecto.2019.228207,
2019b.
Zhang, Z., Xiao, W., Ji, W., Majidifard, M. R., Rezaeian, M., Talebian, M., Xiang,
D., Chen, L., Wan, B., Ao, S., and Esmaeili, R.: Geochemistry, zircon U-Pb
and Hf isotope for granitoids, NW Sanandaj-Sirjan zone, Iran: Implications
for Mesozoic-Cenozoic episodic magmatism during Neo-Tethyan
lithospheric subduction, Gondwana Research, 62, 227-245,
https://doi.org/10.1016/j.gr.2018.04.002, 2018.
Zhao, D., Chen, H., Yang, S., Shen, X., Zhu, G., Li, J., Zhang, G., and Xiao, W.:
Structural style of the foreland fold and thrust belt in the Tianshuihai area,
Western Kunlun, and its tectonic evolution, Acta Geologica Sinica, 74, 134-
141, 2000 (in Chinese with English abstract).
Zhao, G., Wang, Y., Huang, B., Dong, Y., Li, S., Zhang, G., and Yu, S.:
Geological reconstructions of the East Asian blocks: From the breakup of
Rodinia to the assembly of Pangea, Earth-Science Reviews, 186, 262-286,
https://doi.org/10.1016/j.earscirev.2018.10.003, 2018.
Zhao, J., Zeng, X., Tian, J., Hu, C., Wang, D., Yan, Z., Wang, K., and Zhao, X.:
Provenance and paleogeography of the Jurassic Northwestern Qaidam
Basin (NW China): Evidence from sedimentary records and detrital zircon
geochronology, Journal of Asian Earth Sciences, 190, 104060,
https://doi.org/10.1016/j.jseaes.2019.104060, 2020.



Zhao, Z.-F., Dai, L.-Q., and Zheng, Y.-F.: Postcollisional mafic igneous rocks
record crust-mantle interaction during continental deep subduction,
Scientific Reports, 3, 3413, 10.1038/srep03413, 2013.
Zheng, Y., Mao, J., Chen, Y., Sun, W., Ni, P., and Yang, X.: Hydrothermal ore
deposits in collisional orogens, Science Bulletin, 64, 205-212,
https://doi.org/10.1016/j.scib.2019.01.007, 2019.
Zhou, C.-A., Song, S., Allen, M. B., Wang, C., Su, L., and Wang, M.: Post-
collisional mafic magmatism: Insights into orogenic collapse and mantle
modification from North Qaidam collisional belt, NW China, Lithos, 398-
399, 106311, https://doi.org/10.1016/j.lithos.2021.106311, 2021.
Zhou, N., Chen, B., Deng, Z., Sang, M., and Bai, Q.: Discovery and Significance
of Early Jurassic Bimodal Volcanic Rocks in Huoshaoyun, Karakoram,
Geoscience, 33, 990-1002, 10.19657/j.geoscience.1000-8527.2019.05.06,
2019 (in Chinese with English abstract).
Zhu, D.-C., Wang, Q., Cawood, P. A., Zhao, Z.-D., and Mo, X.-X.: Raising the
Gangdese Mountains in southern Tibet, Journal of Geophysical Research:
Solid Earth, 122, 214-223, https://doi.org/10.1002/2016JB013508, 2017.
Zhu, R., Zhao, P., and Zhao, L.: Tectonic evolution and geodynamics of the
Neo-Tethys Ocean, Science China Earth Sciences, 65, 1-24,
10.1007/s11430-021-9845-7, 2022.
Zuza, A. V. and Yin, A.: Balkatach hypothesis: A new model for the evolution of
the Pacific, Tethyan, and Paleo-Asian oceanic domains, Geosphere, 13,
1664-1712, 10.1130/ges01463.1, 2017.





**Supplementary Materials**

Table S1: Analytical methodology.

Table S2: Zircon U-Pb data of Jurassic basalt and sedimentary rocks.

Table S3: Trace element of zircons.

Table S4: Jurassic conglomerate clast lithologies.

Table S5: Whole rock geochemical results of Jurassic basalts.

Fig. S1: Correlations between the trace elements of Jurassic basalts.







Figure 1 (a) Tectonic plate framework in the Northern Hemisphere and the suture zones within the Tethyan Realm (modified from Wu et al., 2020); (b) Structural framework of central Asia showing main blocks and orogenic belts, with locations of major sutures and boundary faults: TFF-Talas-Fergana Fault, BNS-Bangong-Nujiang suture, IYS-Indus-Yalu suture, ATF-Altyn-Tage Fault; (c) Simplified geologic map of the Western Kunlun Mountains including major units and suture zones (modified from Wu et al., 2021; cooling ages of basements refer to Yang et al., 2017): ①- Early Paleozoic Kudi suture, ②- Triassic Mazar-Kangxiwa suture, ③- Triassic Tanymas suture separating the North and Central Pamirs, ④- Rushan-Pshart zone separating the Central and South Pamirs; (d) A section across the east part of the Western Kunlun Mountains showing the deformed and fragmented Jurassic basin. The section location is presented in Fig.1 (b).



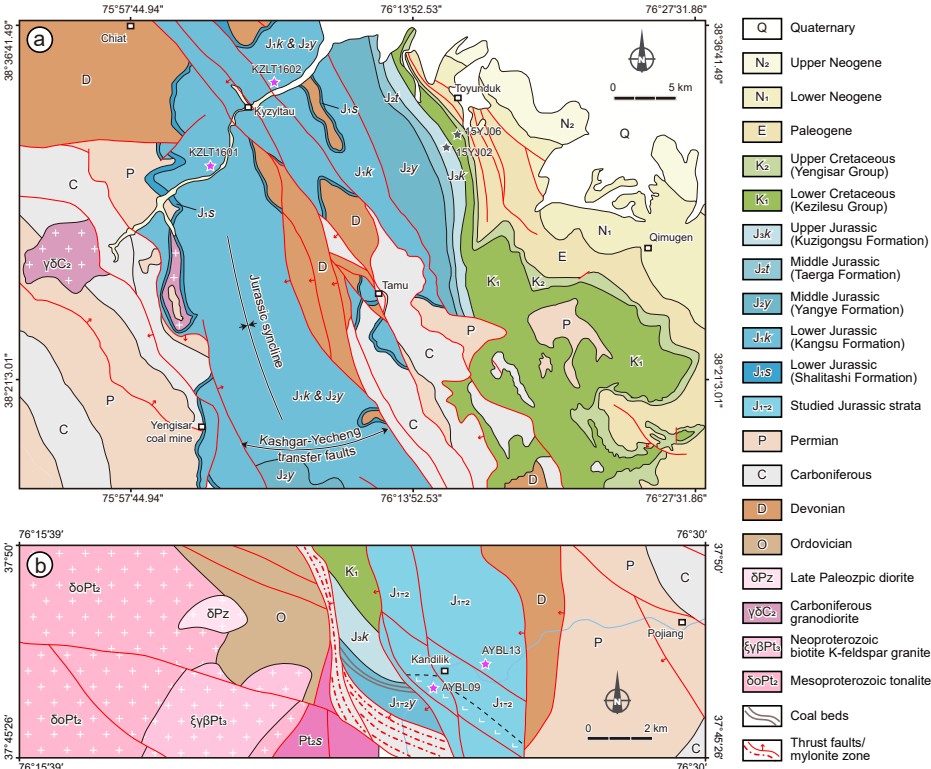

Figure 2 (a) Geological map in the Kyzyltau region showing the stratigraphic information and sampling locations ; (b) Geological map in the Kandilik region showing the Proterozoic basements and Paleozoic-Mesozoic strata. The red stars mark sampling locations in this work, and the grey stars mark the locations of published data (Zhang et al., 2019b).



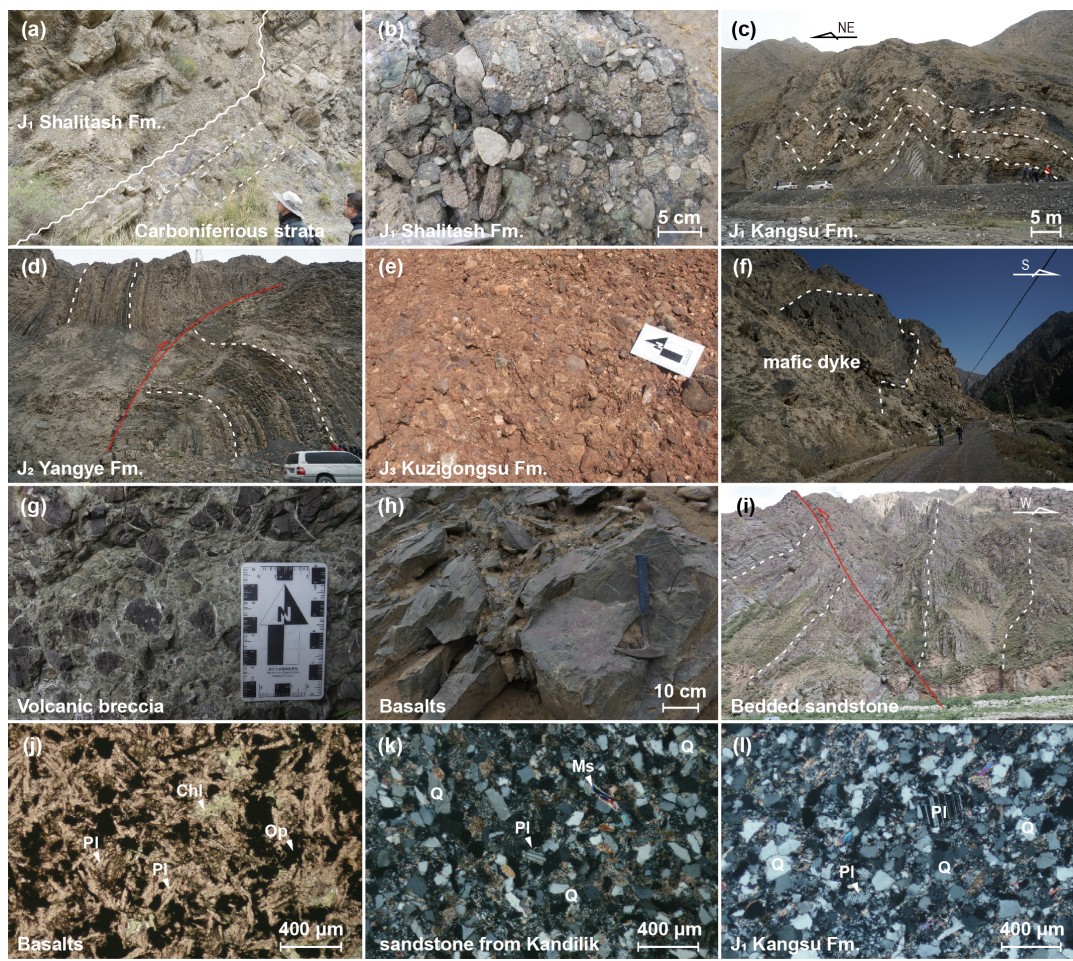

Figure 3 Photographs showing the observation from field and binocular microscope. (a) Early Jurassic Shalitash Formation overlying on the deformed Carboniferous strata with angular unconformity; (b) Conglomerate clast lithologies in the Shalitash Formation; (c) Early Jurassic Kangsu Formation with strongly deformed sandstone layers; (d) Strong deformation of the turbidite sequences in the Middle Jurassic Yangye Formation; (e) Conglomerate clast lithologies in the Late Jurassic Kuzigongsu Formation; (f) Mafic dyke within newly identified Jurassic strata in the Kandilik region; (g) Basaltic volcanic breccia; (h) Massive basalt layer; (i) Jurassic bedded feldspar lithic sandstones with great thickness, which was previously assigned to be Precambrian age; (j) Micrograph of basalt under plane-polarized light; (k) Micrograph of Jurassic sandstone under cross-polarized light from Kandilik section; (l) Micrograph of Jurassic sandstone under cross-polarized light from Kyzyltau section.



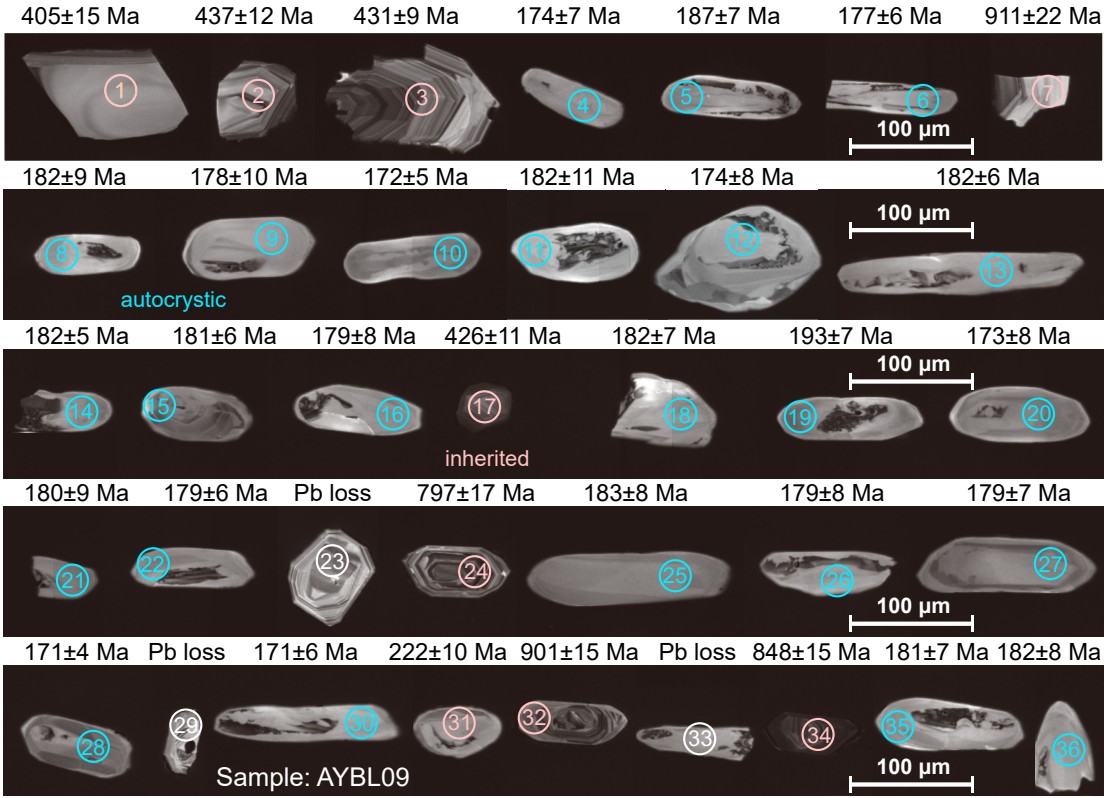

Figure 4 CL images of various kinds of zircons in basalt sample AYBL09, noting the apparent $^{206}$Pb/$^{238}$U ages above.





Figure 5 (a) Concordia plot of LA-ICP-MS U-Pb analysis for the zircons of the basalt sample AYBL09; (b) Weighted mean $^{206}$Pb/$^{238}$U age and concordia age of the youngest zircon groups; (c) Zircon chondrite-normalised REE pattern of the basalt; (d) Th/U ratios of zircons from basalt and sandstone samples. (e) Yb/Sm-Y plotting to distinguish the origins of zircons from the basalt.



Figure 6 Concordia diagram for the detrital zircons of (a) sample AYBL13 from Kandilik section, (c) sample KZLT1601 from Kangsu Formation, and (e) sample KZLT1602 from Yangye Formation; Diagram of the Kernel Density Estimate of detrital zircon U-Pb ages for (b) AYBL13, (d) KZLT1601, and (f) KZLT1602.





Figure 7 Geochemical classification diagram of Jurassic basalt samples from the Kandilik region in the West Kunlun Mountains (green) and from Longshan Formation in the Tianshuihai terrane (blue): (a) total alkali versus silica (Middlemost, 1994) and (b) $Zr/TiO_2$ vs. Nb/Y diagrams (Winchester and Floyd, 1977); (c) Rare earth elements pattern (REE) and (d) trace element diagrams of Jurassic basalts; (e) Th/Yb vs. Nb/Yb plot (Pearce, 2008) and (f) La/Sm vs. La plot (Aldanmaz et al., 2000) Chondrite-normalized REE and the primitive mantle-normalized values refer to Sun and McDonough (1989). The range of the Mariana back-arc basalts refers to Pearce (2008) and the range of eastern China Cenozoic basalts refers to Guo et al. (2020).



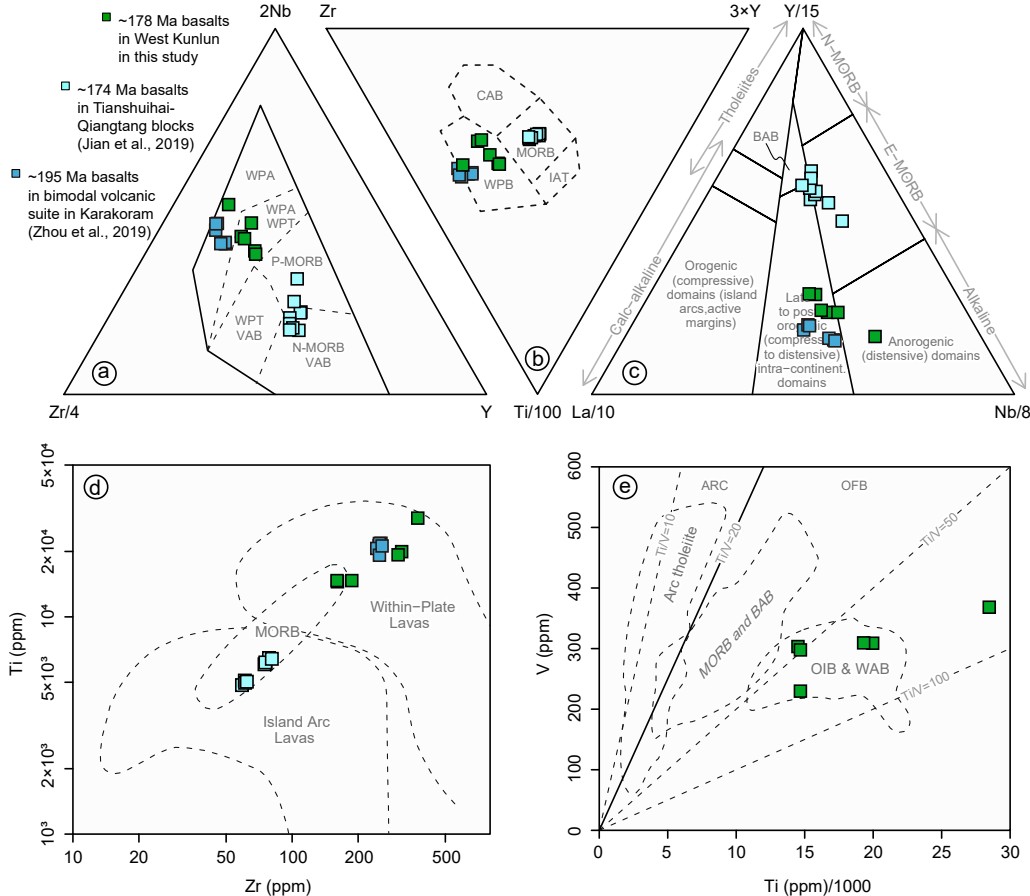

Figure 8 Tectonic discrimination diagrams for Jurassic basalts through (a) Zr/4-2Nb-Y plot and (b) Zr-3Y-Ti/100 plot (Meschede, 1986), (c) La/10-Y/15-Nb/8 plot (Cabanis and Lecolle, 1989), (d) Ti-Zr plot (Pearce, 1982) and (d) V-Ti/1000 plot (Rollinson, 1993). Abbreviation: WPB-within plate basalts; WPA- within plate alkali basalts; WPT-within plate tholeiites; VAB-volcanic arc basalts; CAB- calc-alkali basalts; IAT-island arc tholeiites; BAB-back arc basalts.





Figure 9 Stratigraphic correlations of Jurassic basin along east flank of the Western Kunlun Mountains and the results of gravel analysis of Early Jurassic conglomerate.




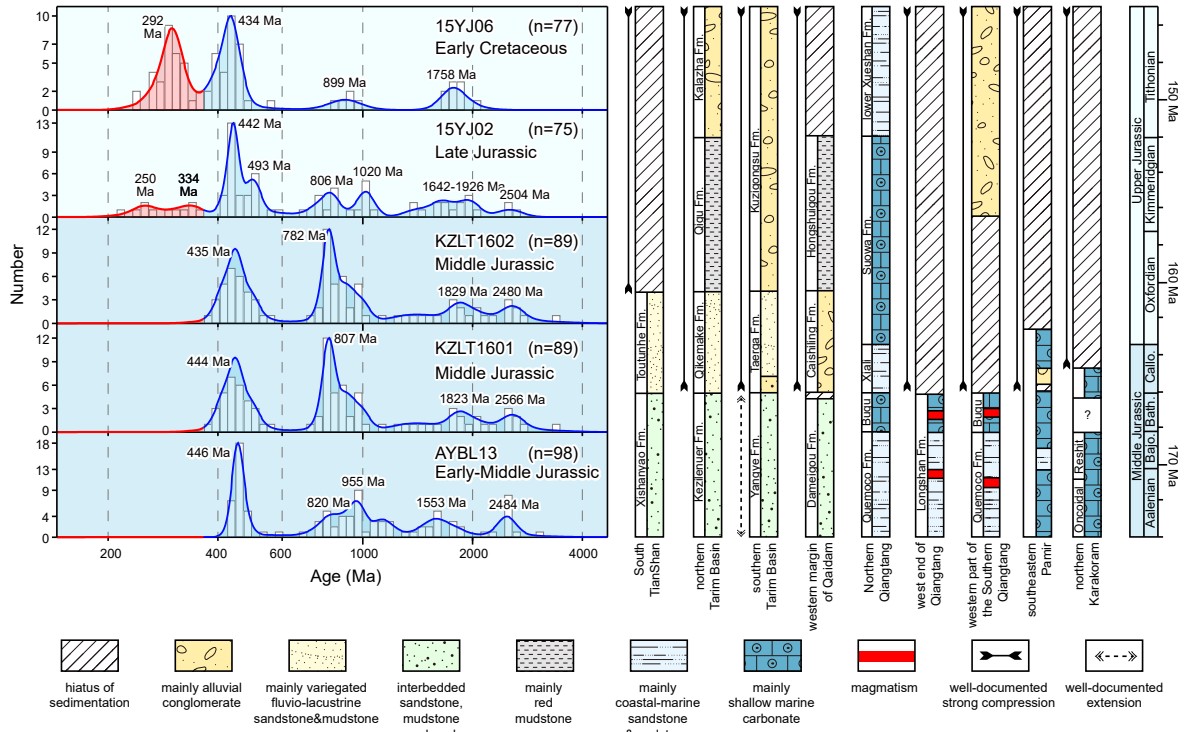

Figure 10 Late Jurassic basin inversion based on the provenance variation through the Early Jurassic to Early Cretaceous and the stratigraphic correlation in the northwestern China. Late Jurassic and Early Cretaceous sandstone samples are according to Zhang et al. (2019b), stratigraphic correlation is modified from Yang et al. (2017).



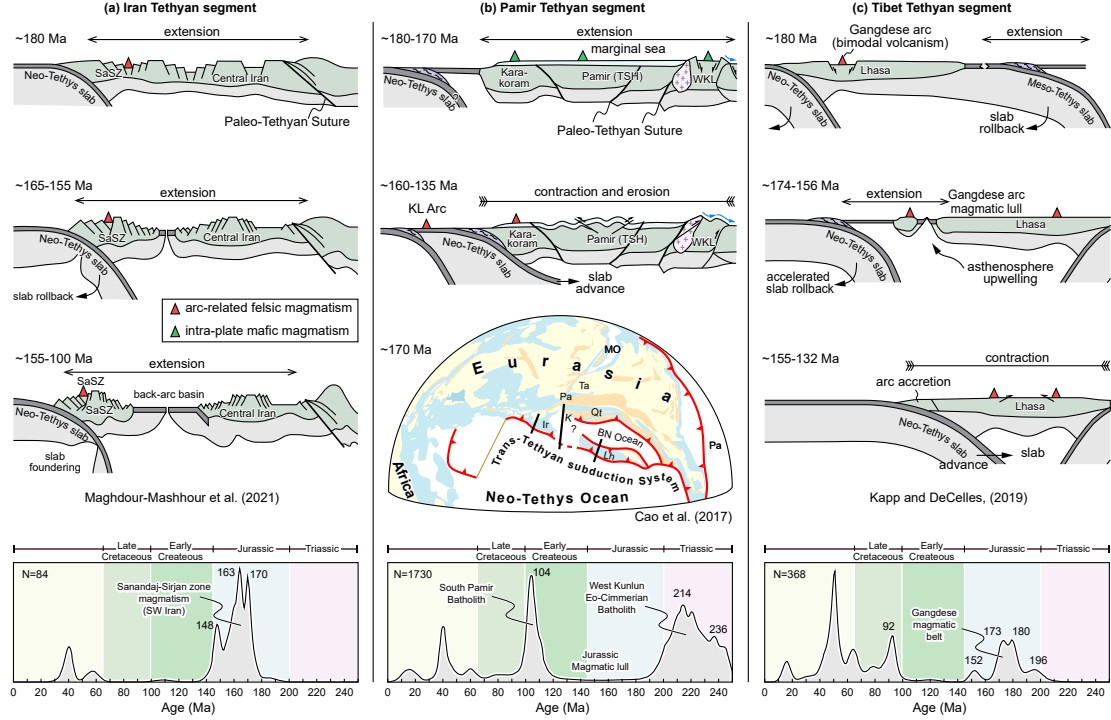

Figure 11 Illustrative cartoons indicating the tectonic variation of the southern Eurasian margin in Jurassic. The subduction of the Neo-Tethys Ocean resulted in persistent rifting along the Iran Tethyan segment, generating massive magmatism during the Early Jurassic to Early Cretaceous. The far-field subduction causing the Early-Middle Jurassic extension along the Pamir Tethyan segment without magmatic flare-up. The changes in subduction style along the Pamir and Tibet Tethyan segments induced the extension-contraction transition. The spatial magmatic datasets are according to Zhang et al. (2018), Chapman et al. (2018), Ma et al. (2017) and Zhu et al. (2017), and the map of paleogeographic reconstruction is modified from Cao et al. (2017). Abbreviation: SaSZ-Sanandaj-Sirjan zone; TSH-Tianshuihai block; WKL-West Kunkun Mountains; KL Arc- Kohistan Ladakh Arc; Ir-Iran; K-Karakoram; Pa-Pamir; Ta-Tarim; Qt-Qiangtang; Lh-Lhasa; BN Ocean-Bangong-Nujiang Ocean.