# Peer review of "Switching Extensional and Contractional Tectonics in the 1 West Kunlun Mountains During Jurassic: Responses to the 2 Neo-Tethyan Geodynamics along the Eurasian Margin 4 Hong-Xiang Wu1,2, Han-Lin Chen1,2\*, Andrew V. Zuza3, Yildirim Dilek4, Du-We"

_EGUsphere, 2024_

## Author Response (AR1)

**Response letter for EGUSPHERE-2024-1670**

**Response to comments from anonymous referee #1:**

**Comment:** This work reports field-based, geochronological and geochemical data for the Jurassic volcaniclastic sequences in West Kunlun. New results provide constraints on the Mesozoic tectonic history of the central junction of the Tethyan Orogenic Belt, of importance to understand the Tethyan geodynamics. The data is of good quality and the figures are well presented. Minor revision is suggested before publication, especially regarding age and tectonic setting explanations. English also needs polishing. My major and minor comments are presented in the following.

**Author's Response:** Thank you very much for your time and valuable suggestions. We will address your queries individually below. Additionally, our co-authors, two American scholars, Yildrim and Andrew, have contributed to revising and refining the language.

**Major comments**

*Comment 1:* "Alternating" in the title may not be quite appropriate, as it means multiple changes. But the main finding of this work is a transition?

*Author's response:* We agree with your suggestion.

*Author's change:* We have revised our title to: 'Switching Extensional and Contractional Tectonics in the West Kunlun Mountains During Jurassic.'

*Comment 2:* The section 2 concerning geological framework is supposed not to describe too many debates (Lines 190-207).

*Author's response and change:* We have removed the discussion regarding the interpretation of Jurassic molasse deposits; please refer to line 191. We suggest retaining the first sentence in this paragraph, as the first-order geodynamic mechanisms in the mid-Mesozoic are the most important focus of this paper.

*Comment 3:* Is there any corresponding relationship among morphology, internal structure under CL, and age of zircons from the basalt sample? These zircons were sub-categorized into two groups based on the presence of oscillatory zoning, but whether the two group zircons have distinctive ages and Th/U ratios have not been clearly mentioned. Besides, the zircons from basalt are not supposed to have oscillatory zoning, which is a typical feature of zircons from felsic magma. Please reconsider the above-mentioned issues and add more discussion.

*Author's response:* Based on a detailed classification and statistical analysis of zircon characteristics in Table S2, we identified a strong correlation among the morphology, internal structure, and age of zircons from the basalt. Type 1 zircons, typically euhedral to sub-euhedral with clear oscillatory zoning, have older ages ranging from 405 Ma to 911 Ma. In contrast, type 2 zircons, which exhibit subrounded external shapes, show uniform Jurassic ages between 168 Ma and 193 Ma.

We have assigned the younger ages (type 2) to represent the crystallization age of this basaltic rock sample, while the older ages (type 1) are

interpreted as inherited from the country rocks. Notably, the older ages in the basalt sample are consistent with the detrital zircon ages found in the study region.

**Author's change:** Please refer to discussion in line 337-342 in our revised manuscript.

**Comment 4:** Why is there a significant gap between the youngest zircon age and the depositional age of the sedimentary rocks? Except for the age of basalt, is there any other evidence to indicate their Jurassic depositional age?

**Author's response:** Firstly, the provenance of the Jurassic sedimentary rocks is locally sourced from the North West Kunlun Mountains, which are primarily composed of Early Paleozoic sedimentary rocks (lines 616-621). Younger detrital zircons, dating of the Carboniferous and Triassic ages, in the South West Kunlun Mountains, were transported into the Tarim Basin beginning in the Late Jurassic and Early Cretaceous (Fig. 10). As a result, a significant gap between the youngest zircon age and the actual depositional age can be observed.

Secondly, the Early Jurassic basalts belong to the upper member of strata deposited above the thick clastic strata based on a previous stratigraphic measurement (Ma et al., 1991). Accordingly, late magmatic rocks are unable to provide debris to earlier formations. Additionally, the Early Jurassic basalts are scattered throughout this region, making them insufficient to serve as a major source. Zircon is also difficult to crystallize in these mafic magmas due to their silicon unsaturation.

To better understand the Jurassic depositional age of this sedimentary

package, we present several lines of evidence as follows: (1) the age spectrum of the sandstone from Kandilik is remarkably similar to that of the Early Jurassic strata (Fig. 6); (2) the structural compatibility of this new stratigraphic scheme was we demonstrated (line 489-501); (3) The clastic member in Kandilik is primarily composed of gray-black carbonaceous slate and siltstone, similar to the Jurassic coal-bearing sequences.

**Comment 5:** The length of the text is suggested to be largely reduced. For example, there are some overlaps between 6.1 concerning the tectonic setting of the Jurassic volcanism and 6.2 about the setting under which the Jurassic basin formed.

**Author's response:** We agree with your suggestion.

**Author's changes:** We have significantly shortened the discussion about the tectonic setting in the section 6.1 and 6.2. Please refer to line 548-574.

**Comment 6:** 6.3 is discussed largely based on previous work. This contribution is suggested to be highlighted in this section. Besides, better add some summary sentences to conclude the main findings.

**Author's response:** We briefly discussed the comparable geological history between the Pamir-West Kunlun and the Tibetan Plateau and added a summary of this in the conclusion section.

**Minor comments**

**Comment 1:** Better add the columns or sections of the Jurassic strata in the studied area in Figure 2.

*Author's response:* We have added a field geological section to show the regional strata and deformation in Figure 2c.

*Comment 2:* Better add some photos to show the different clast compositions of Jurassic conglomerate.

*Author's response:* We have added several field photos of conglomerate clast lithologies from Oytag, Gaizi and Tamu regions in Figure 2.

*Comment 3:* Figure 4: the various colored circles are suggested to be annotated to represent different age explanations.

*Author's response:* We have explained the meaning of the different colored circles in the caption of Figure 4.

*Comment 4:* Figure 5b: what are dots 19, 25, 28, and 30?

*Author's response:* The subscale numbers in Figure 5b represent the zircon test sites shown in Figure 5a that were not used to calculate the weighted mean age. We have deleted these subscale numbers in this revised version.

*Comment 5:* The ~195 Ma basalts in the bimodal volcanicsuite in Karakoram are plotted in Figures 7-8, but comparison has not been made with the studied samples. What is their significance? Better add some discussion.

*Author's response:* We have highlighted the chemical differences among magmatic rocks in the discussion section 6.3 (lines 687-692). In particular, we emphasize the back-arc MORB affinity of the basalts in Tanshuihai and the OIB-affinity (within-plate) of the basalts in West Kunlun, which is comparable to the

tectonic setting of the active margin of the western Pacific (line 715-720).

**Comment 6:** Line 517: slightly negative

**Author's response:** We have corrected this mistake.

**Comment 7:** Lines 587-592: lowercase following (1), (2), (3), and (4).

**Author's response:** We have corrected this mistake.

**Comment 8:** Condense 6.2 and add some summary sentences for this section.

**Author's response:** We have shortened both the section 6.1 and 6.2 in this revised manuscript, and have added a summary of basin evolution for the section 6.2 (line 642-647).

**Response to comments from anonymous referee #2:**

**Comment:** The authors investigate the Jurassic basins in the West Kunlun Mountains. Based on the presence of oceanic island basalts and an upward-fining sedimentary pattern, they argue for an extensional setting in the Early to Middle Jurassic in this region, before basin inversion occurred in the Late Jurassic. They propose that the basin evolution was related to the northward subduction of the Neo-Tethys Ocean. Overall, I find this study data-rich and well-written, and it should be published after minor revisions. Below, please find several comments that I hope may be useful for the authors to improve the manuscript.

**Author's response:** Thank you very much for your consideration and suggestion. We will address your questions and make the following revision.

**Comment 1:** Lines 141-142: The South Qiangtang and North Qiangtang terranes are larger than most other terranes you mentioned. They should not be omitted here. Additionally, some authors believe North Qiangtang had a Cathaysian affinity.

*Author's response:* We have mentioned the South Qiangtang and North Qiangtang terranes in this paragraph.

*Author's change:* please refer to the revision in line 141.

**Comment 2:** Figure 3h: No layers can be identified. Also, in which strata are these basalts found?

*Author's response:* Due to the exposure conditions and harsh terrain, we

were unfortunately unable to capture clear photographs showing the contact between the basalt and surrounding strata. These basalts were found in the upper layer of volcanic breccia, and detailed profiles were measured by geologists as early as the 1990s (Ma et al., 1991).

*Author's change:* To provide readers with a clearer understanding of the strata formation and rocks we collected, we have added a field outcrop profile of the area in Fig. 2c.

**Comment 3:** Figure 5: I have concerns about the zircons. From the CL image, they appear to originate from felsic rocks. Additionally, there are multiple age clusters, which seems unusual to me. Also, how do you explain the Th/U values of zircons being lower than 0.1?

*Author's response:* In this revised version, we have conducted a detail classification and statistical analysis of zircon characteristics in Table S2. Based on the morphology, internal structure (CL), and age of zircons from the basalt, we have divided the 36 zircons we tested into two groups. The type 1 zircons, typically euhedral to sub-euhedral with clear oscillatory zoning, have older ages ranging from 405 Ma to 911 Ma. In contrast, the type 2 zircons, exhibiting subrounded external shapes with no clear oscillatory zoning, show uniform Jurassic ages between 168 Ma and 193 Ma.

The older ages in the basalt sample are consistent with the detrital zircon ages found in the country rocks in the study region (Fig. 6). Accordingly, we have assigned the older ages (type 1) as inherited from the country rocks, while he younger ages (type 2) to represent the crystallization age of this basaltic rock sample.

The low Th/U ratios, occurred only within the zircon group of type 2, which has a "polished" shape with nebulous or patchy-zoned centers. This may result from moderate resorption either during the evolution of the magma chamber when the magma is oversaturated with respect to zircon or a certain degree of metamorphism (Corfu et al., 2003). Despite this, most zircons still display high Th/U ratios, indicating a clear magmatic origin. Therefore, we believe the Jurassic age represents the crystallization age of the volcanic rocks.

*Author's change:* We have revised our description of the results. Please refer to discussion in line 337-342.

*Comment 4:* The lack of Jurassic and the presence of very few Triassic detrital zircons in the Jurassic strata, with a strong 440 Ma peak, suggests there was no widespread arc magmatism in the region during the Late Silurian to Triassic. Therefore, detrital zircon ages may not be that useful for determining depositional ages. Do you have any other evidence for the depositional ages?

*Author's response:* To better understand the Jurassic depositional age of this sedimentary package, we present several lines of evidence as follows: (1) the age spectrum of the sandstone from Kandilik is remarkably similar to that of the Early Jurassic strata (Fig. 6); (2) the structural compatibility of this new stratigraphic scheme was we demonstrated (line 489-501); (3) The clastic member in Kandilik is primarily composed of gray-black carbonaceous slate and siltstone, similar to the Jurassic coal-bearing sequences.

*Comment 5:* The Pamirs can be subdivided further and then correlated with Tibet. Although there is considerable debate on the correlation due to the

presence of the Karakorum fault, the authors should not overlook this in their discussion. They should consider the role of the Meso-Tethys Ocean in Tibet and the Pamirs in more detail. From what I know of the Tibetan part, the Jurassic was an important period during which the Bangong-Nujiang Meso-Tethys Ocean was subducting and experiencing microcontinent assemblage (Ma et al., 2023, Tectonophysics 862, 229957). The authors mentioned the regional unconformity beneath the Late Jurassic conglomerate, which is important, as nearly synchronous unconformities also exist in the South Qiangtang and Bangong-Nujiang suture zones (Ma et al., 2017, Journal of Geophysical Research: Solid Earth 122(7), 4790-4813; 2018, Palaeogeography, Palaeoclimatology, Palaeoecology 506, 30-47). It would be interesting if the authors could provide more details about the basin inversion and further discuss the relationship with Meso-Tethys geodynamics.

*Author's response:* We have added more information about this event in the South Qiangtang and Bangong-Nujiang suture zone. We also cited these important papers in our revised manuscript (line 233 and 626). In the final section, we further discussed the relationship between the basin inversion in South Qiangtang and the evolution of the Meso-Tethys.

*Comment 6:* Figure 10: I suggest adding the log of the West Kunlun here for comparison.

*Author's response:* Jurassic strata are entirely absent in the West Kunlun Mountains. Only in some region, such as east of Tashkurghan, the Lower Cretaceous reddish sandstones unconformably overlie Paleozoic strata and Triassic granitoids.

**References**

Ma, S., Wang, Y., and Fang, X.: Basic characteristics of Proterozoic Eonothem as a table cover on northern slope, Xinjiang Geology, 9, 59-71, 1991 (in Chinese with English abstract).

Corfu, F., Hanchar, J. M., Hoskin, P. W. O., and Kinny, P.: Atlas of Zircon Textures, Reviews in Mineralogy and Geochemistry, 53, 469-500, 10.2113/0530469, 2003.

---

## Author Response (AR2)

**Response letter for EGUSPHERE-2024-1670**

**Responses to Editor**

**Editor:** Along with some suggestions by the reviewers, this manuscript only needs several corrections. Some words are loosely used and deserve more attention. Below are the details:

**Author:** Thank you for your decision and detailed comments. We have thoroughly reviewed the issues raised and made the necessary revisions accordingly.

**Editor:** Line 27 (tracked version): Western Asia normally refers to Iran to Turkey. This location is a little confusing.

**Author:** We have changed Western Asia to Pamir orogens.

**Editor:** Line 39: Change "integrating...with..." to "including...and..."

**Author:** We have revised accordingly.

**Editor:** Line 45: Which basin? The abstract does not mention it.

**Author:** We have indicated the southwestern Tarim basin in this version.

**Editor:** Line 58-59: The Tethyan Orogenic Belt is an over 15,000 km trans-Eurasian orogenic system with a series of mountain chains and orogenic plateaus.

**Author:** We have revised accordingly.

**Editor:** Line 64: Closure.

**Author:** We have revised accordingly.

**Editor:** Line 69: Delete "the".

**Author:** We have revised accordingly.

**Editor:** Line 75: Delete "a protracted phase of".

**Author:** We have revised accordingly.

**Editor:** Line 79-80: The Mesozoic tectonic evolution of the Tethyan realm exhibits···

**Author:** We have revised accordingly.

**Editor:** Line 82: Diverse->different

**Author:** We have revised accordingly.

**Editor:** Line 83: What is the meaning of "a propagating continental rift system"? Unclear. Reword it.

**Author:** We have made an explanation as a northwestward-propagating continental rift system.

**Editor:** Line 84: Central Iran Block

**Author:** We have revised accordingly.

**Editor:** Line 107: Critically represents? Records.

**Author:** We have changed the expression.

**Editor:** Line 128: This basin is mysterious.

**Author:** We have detailed this as the 'Jurassic Kyzyltau-Kandilik basin'.

**Editor:** Line 131: and probes···

**Author:** We have revised accordingly.

**Editor:** Line 137: Delete "is a vast, east-west-extending mountain system that". Repeated.

**Author:** We have revised accordingly.

**Editor:** Line 177: Deflected? I can only imagine displaced mountains by the Altyn tagh Fault.

**Author:** We have used displaced instead of deflected.

**Editor:** Line 184: Delete "floor".

**Author:** We have revised accordingly.

**Editor:** Line 188: Remove "much later"

**Author:** We have revised accordingly.

**Editor:** Line 191: Integration->amalgamation or accretion

**Author:** We have revised as amalgamation.

**Editor:** Line 196: Delete observed

**Author:** We have revised accordingly.

**Editor:** Line 195: Remove plutons or granitoids.

**Author:** We have deleted plutons.

**Editor:** Line 201-203: A Jurassic to Cretaceous polymetamorphic history is also displayed by monazite ages.

**Author:** We have revised accordingly.

**Editor:** Line 206: High-flux event? Not clear.

**Author:** We have reworded the expression.

**Editor:** Line 215: Using "thickest" should have a limit in area.

**Author:** We have used thick.

**Editor:** Line 239-244: Delete this sentence irrelevant to the geological setting.

**Author:** We have removed these sentences.

**Editor:** Line 249-250: A slate unit has unmetamorphosed sandstone and siltstone. This is nonsense.

**Author:** We have rephrased this sentence.

**Editor:** Line 268: Which are these oxides? There is no explanation for abbreviations in fig. 3m.

**Author:** We have added the mineral abbreviation in the caption of Figure 3.

**Editor:** Line 288: Which section?

**Author:** We have indicated this section as Kandilik section in line 280.

**Editor:** Line 350: Delete "of the zircons"

**Author:** We have revised accordingly.

**Editor:** Line 392: Delete age.

**Author:** We have revised accordingly.

**Editor:** Line 400: Delete revealed.

**Author:** We have deleted it.

**Editor:** Line 403: Delete regional detrital.

**Author:** We have deleted it.

**Editor:** Line 425: What is green sandstone? Delete "green" for sandstone here and elsewhere.

**Author:** The green-colored and reddish sandstones are the typical sediments of regional Devonian strata, which implicated the potential provenance.

**Editor:** Line 428: Delete variegated.

**Author:** We have deleted it.

**Editor:** Line 437: Delete green-colored.

**Author:** We have indicated that the green-colored sandstones are important marker for the source of Jurassic conglomerate.

**Editor:** Line 446: This section must follow the age of the basalts, not here.

**Author:** We have replaced this section below the section 4.1 (line 354).

**Editor:** Line 482: Delete identified.

**Author:** We have deleted it.

**Editor:** Line 490: Delete are.

**Author:** We have deleted it.

**Editor:** Line 494: Delete monotonous

**Author:** We have deleted it.

**Editor:** Line 497: What is lithological makeup? Reword it.

**Author:** We have reworded it as composition.

**Editor:** Line 503: Delete "of the youngest group of zircons separated"

**Author:** We have deleted it.

**Editor:** Line 518: Delete exposed.

**Author:** We have deleted it.

**Editor:** Line 523: Delete dominated.

**Author:** We have deleted it.

**Editor:** Line 524: If it is not exhumed, how could it become a provenance?

**Author:** We have deleted exhumed.

**Editor:** Line 526: Delete structural.

**Author:** We have deleted it.

**Editor:** Line 528-532: Why? Show some evidence of turbidite here.

**Author:** We have rephrased this sentence.

**Editor:** Line 490: Hereon?

**Author:** We have deleted it.

**Editor:** Line 535-538: Rephrase this sentence.

**Author:** We have rephrased this sentence.

**Editor:** Line 539-542: I do not understand this tectonic relationship. Try to make it clear.

**Author:** We have rephrased this sentence.

**Editor:** Line 542: These two units successfully extend into ⋯ Awkward expressions.

**Author:** We have rephrased this sentence.

**Editor:** Line 545: Delete of the.

**Author:** We have deleted it.

**Editor:** Line 550: Tectonic setting of the Early Jurassic volcanism. Besides, this section repeats the geochemistry description of basalts, that should be removed.

**Author:** We have revised the title of this section. Additionally, we have removed the repeated geochemistry description in the first paragraph.

**Editor:** Line 568: Negligible crustal contamination. Delete "during their journey to the surface"

**Author:** We have revised this sentence accordingly.

**Editor:** Line 575: Is this volcanism 178 Ma?

**Author:** We have added the age information (178 Ma) in this sentence to make it more understandable.

**Editor:** Line 576: Delete regionally.

**Author:** We have deleted it.

**Editor:** Line 583: Extrusion? Or mixing?

**Author:** We have reworded this sentence.

**Editor:** Line 584-590: This part is unclear. Delete it.

**Author:** We have removed these sentences.

**Editor:** Line 593: Name the type of rocks instead unclear types (post/syn-orogenic).

**Author:** We have indicated the rock types.

**Editor:** Line 595-604: Determine basalt setting with details of elements, rather than simply saying "plots indicate…"

**Author:** We have revised the discussion of the tectonic setting for the basalt by adding more details of elemental information.

**Editor:** Line 615-619: Give some structural evidence in this region and then add this along-strike comparison to support the inference.

**Author:** We have added more structural evidence before the model proposed.

**Editor:** Line 628-635: Rewrite it to cope with the following paragraphs.

**Author:** We have rephrased the first paragraph.

**Editor:** Line 577: Along-strike.

**Author:** We have revised accordingly.

**Editor:** Line 651-659: The first sentence and fig. 9 better demonstrate the idea. The rest is less necessary.

**Author:** We have removed these useless sentences, and now it looks better.

**Editor:** Line 593: In contrast

**Author:** We have removed this part.

**Editor:** Line 702: Replace "creating an elevated topography, which in turn" with "and".

**Author:** We have rephrased this sentence.

**Editor:** Line 705-706: A basin on an unconformity? Unclear description.

**Author:** We have rephrased this sentence.

**Editor:** Line 655: Seaways->oceans

**Author:** We have used oceans instead.

**Editor:** Line 702-723: Delete this sentence. Unnecessary.

**Author:** We have removed these sentences.

**Editor:** Line 725-728: This importance has been emphasized before.

**Author:** We have removed these repeated sentences.

**Editor:** Line 745: Delete there.

**Author:** We have deleted there.

**Editor:** Line 746: Delete "Available geochronological data indicate that"

**Author:** We have deleted.

**Editor:** Line 750: Delete "Geochemical studies reveal that"

**Author:** We have deleted it.

**Editor:** Line 766: Delete expressed

**Author:** We have deleted it.

**Responses to reviewer**

**Editor:** Most comments have been addressed, and there remains minor clarification before publication.

**Author:** Thank you for your careful review and for providing constructive feedback. In this updated version of the manuscript, we have significantly reduced the overlapping content in the regional background and discussion sections, in line with the editor's suggestions. This has improved the readability of the article. We believe this also addresses, to a large extent, your earlier suggestion to substantially shorten sections 6.1 and 6.2. Additionally, we have made smoother adjustments to the results section.